



# A change in perspective: Downhole cosmic-ray neutron sensing for the estimation of soil moisture

Daniel Rasche[1,*], Jannis Weimar[2,*], Martin Schrön[3], Markus Köhli[2,4], Markus Morgner[1],
Andreas Güntner[1,5], and Theresa Blume[1]

[1]GFZ German Research Centre for Geosciences, Section Hydrology, 14473, Potsdam, Germany
[2]Physikalisches Institut, Heidelberg University, Im Neuenheimer Feld 226, 69120 Heidelberg, Germany
[3]UFZ – Helmholtz Centre for Environmental Research GmbH, Dep. Monitoring and Exploration Technologies, Permoserstr. 15, 04318, Leipzig, Germany
[4]Physikalisches Institut, University of Bonn, Nussallee 12, 53115 Bonn, Germany
[5]University of Potsdam, Institute of Environmental Sciences and Geography, 14476, Potsdam, Germany

[*]Daniel Rasche and Jannis Weimar contributed equally to this work.

**Correspondence:** Daniel Rasche (daniel.rasche@gfz-potsdam.de), Jannis Weimar (weimar@physi.uni-heidelberg.de)

**Abstract.**

Above-ground cosmic-ray neutron sensing (CRNS) allows for the non-invasive estimation of the field-scale soil moisture content in the upper decimetres of the soil. However, large parts of the deeper vadose zone remain outside of its observational window. Retrieving soil moisture information from these deeper layers requires extrapolation, modelling, or other methods, all of which come with methodological challenges. Against this background, we investigate CRNS for downhole soil moisture measurements in deeper layers of the vadose zone. To render calibration with in-situ soil moisture measurements unnecessary, we re-scaled neutron intensities observed below the terrain surface with intensities measured above a water body.

An experimental set-up with a CRNS sensor deployed at different depths up to 10 meters below the surface in a groundwater observation well combined with particle transport simulations revealed the response of downhole thermal neutron intensities to changes in soil moisture content at the depth of the downhole neutron detector as well as in the layers above it. The simulation results suggest that the sensitive measurement radius of several decimeters, which depends on soil moisture and soil bulk density, exceeds the one of a standard active neutron probe which is only about 30 cm. We derived transfer functions to estimate downhole neutron signals from soil moisture information and we describe approaches for using these transfer functions in an inverse way to derive soil moisture from the observed neutron signals. The in-situ neutron and soil moisture observations confirm the applicability of these functions and prove the concept of passive downhole soil moisture estimation even at larger depths using cosmic-ray neutron sensing.





# 1   Introduction

Soil moisture is a key variable in the hydrological cycle (Vereecken et al., 2008, 2014; Seneviratne et al., 2010) as it drives
energy as well as water fluxes and thereby influences groundwater recharge, runoff generation processes and hence, the local
water balance. It influences vegetation growth and vegetation communities which, in turn, influence the local soil moisture
and microclimate  (e.g. see, Daly and Porporato, 2005; Seneviratne et al., 2010; Wang et al., 2018). Averaged over several
ecosystems approximately 75 percent of roots can be found in the upper 40 cm of the soil (Jackson et al., 1996). As a result
soil moisture in these upper decimeters of the root zone exerts an important control on the hydrological cycle. However,
the maximum rooting depth largely exceeds the upper decimeters of the soil, depends on the plant species (Canadell et al.,
1996) and is driven by local hydrological conditions (e.g. Fan et al., 2017). These deep roots can be of high importance for
water supply of other, more shallow rooting plants through e.g. the process of hydraulic lift (e.g. Neumann and Cardon, 2012;
Pierret et al., 2016), especially during dry periods. Additionally, infiltrating water can be diverted along deep roots to greater
depths as preferential flow (e.g. see, Nimmo, 2021) potentially leading to an increased water storage in deeper layers of the
unsaturated zone. Among others, these processes make deeper layers likewise important for the local water balance and the
local hydrological processes.

As soil moisture is highly variable even on small horizontal scales (Vereecken et al., 2014), a large number of of point-scale
measurements (e.g. in-situ sensors) is required to overcome the small-scale variability and derive representative averages. One
method to directly measure representative soil moisture averages over several hectares and is Cosmic-Ray Neutron Sensing
(CRNS) (Schrön et al., 2018). It has been introduced about a decade ago by Zreda et al. (2008) and Desilets et al. (2010)
and uses secondary neutrons produced from cosmic rays which are inversely correlated with the amount of hydrogen in the
surrounding area. It allows for the non-invasive estimation of average soil moisture contents up to depths of 15 to 83 cm (Köhli
et al., 2015), thus largely covering the shallow soil layers with high root densities.

Despite the large horizontal measurement footprint radius of 130 to 240 m (Köhli et al., 2015), CRNS lacks the integration
depth large enough to cover greater parts of the deeper root zone. Other geophysical methods with a large (hectometer scale)
horizontal measurement area such as geoelectric approaches (Cimpoiaşu et al., 2020; de Jong et al., 2020) and the observation
of integral mass changes by terrestrial gravimetry (Van Camp et al., 2017; Reich et al., 2021) may allow for inferring soil
moisture dynamics in larger depths of the vadose zone. However, the separation of the integral gravity signal into different
hydrological signatures can be challenging (Van Camp et al., 2017). In addition, depending on the geophysical method chosen,
continuous measurements may not be feasible, which would hamper the monitoring of the soil moisture dynamics in the deeper
vadose zone.

Another soil moisture measurement technique with a measurement volume smaller than other geophysical methods but
larger than that of point-scale sensors is the active neutron probe. Invented in the mid of the previous century (Gardner and
Kirkham, 1952), the active neutron probe allows for the estimation of soil moisture in the depth of interest through access-
tubes. Instead of passively observing the flux of naturally occurring epithermal (0.25 eV to 100 keV) neutrons as it is the case of
above-ground CRNS, an active neutron source produces fast neutrons (100 keV to 10 MeV) and a collocated neutron detector



observes the intensity of backscattered slowed-down thermal (below 0.25 eV) neutrons. The intensity of thermal neutrons measured under radiation of a fast neutron source largely depends on the hydrogen content of the soil due to the decelerating power of hydrogen through elastic collisions and removal of thermal neutrons by absorption (see e.g., IAEA - International
Atomic Energy Agency, 1970; Gardner, 1986; Kramer et al., 1992; Ferronsky, 2015, for a detailed review).

An important advantage of the downhole soil moisture estimation using active neutron probes is their decimeter-scale measurement volume around the probe in the soil. Ølgaard's (1965) equation in Kristensen (1973) and Gardner (1986) defines the measurement radius in a surrounding soil volume as the radius $R_{95}$ within which 95 percent of the detected thermal neutron signal originates from. Accordingly, the radius inversely depends on the soil water content, described as $R_{95} \approx 53\,\mathrm{cm}$ for
$\theta = 0.05\,\mathrm{cm}^3\,\mathrm{cm}^{-3}$, or $R_{95} \approx 20\,\mathrm{cm}$ for $\theta = 0.35\,\mathrm{cm}^3\,\mathrm{cm}^{-3}$. As a consequence, the measurement volume of the active neutron probe exceeds the integration volume of standard in-situ point-scale sensors and allow for a more representative average soil moisture value in the depth of interest. However, the disadvantages are the higher precautions that need to be taken when handling active radiation sources (e.g., IAEA - International Atomic Energy Agency, 1970; Gardner, 1986) as well as the typically non-continuous nature of snapshot measurement campaigns with active neutron probes.

Against this background, we investigate the possibility of using CRNS-related sensors in a passive downhole technique (d-CRNS) to estimate soil moisture in different depths of the root zone and deeper unsaturated zone. For this, we installed components of CRNS neutron detectors in a standard groundwater observation well, and thus using the well casing above the groundwater level as an access tube.

We hypothesise that a sufficient neutron intensity can be observed by a downhole neutron detector in order to measure
neutron intensity changes caused by soil moisture dynamics in discrete depths of the soil. Hereby, taking advantage of both, the passive, non-invasive characteristics, and continuous monitoring capabilities of CRNS, as well as the decimeter-scale measurement volume of sub-surface active neutron probes.

Using existing standard groundwater observation wells opens the perspective of a multi-purpose use of existing observational infrastructure as simultaneous groundwater level measurements remain undisturbed.

To test our hypotheses, we first conducted particle transport simulations using the MCNP particle transport code commonly used in CRNS research (e.g., Zreda et al., 2008; Franz et al., 2012; Andreasen et al., 2016, 2017; Weimar et al., 2020; Köhli et al., 2021, among others) to investigate the neutron flux in different soil depths. As we expect the neutron response to changes of moisture in the surrounding soil to be different compared to above-ground CRNS or the active neutron probe, we use the particle transport simulations to obtain information on the integration volume and to derive a transfer function from soil
moisture to neutron intensities. In a second step, we compare the estimated neutron intensities calculated from reference soil moisture observations based on the derived transfer function with measurements of downhole neutron intensities in different depths. Finally, we illustrate the potential of passive downhole Cosmic-Ray Neutron Sensing for the estimation of soil moisture in the vadose zone.





## 2 Material and methods

### 2.1 Study site


The study site comprises the permanent CRNS observation site „Serrahn" (Bogena et al., 2022) which is located in the Müritz National Park in the lowlands of north-eastern Germany (Fig. 1). The site is of one of three permanently operating CRNS stations (Heidbüchel et al., 2016; Rasche et al., 2021) in the Terrestrial Environmental Observatory TERENO-NE (Zacharias et al., 2011; Heinrich et al., 2018). The observatory is located in the cfb climatic zone following the Köppen-Geiger classifica-

tion (Bogena et al., 2022) with an average annual temperature of 8.8°C and precipitation sum of 591 mm per year at the closest long-term weather station in Waren (in a distance of approximately 35 km) operated by the German Weather Service (station ID: 5349, period 1981–2010) (DWD - German Weather Service, 2020a, b).

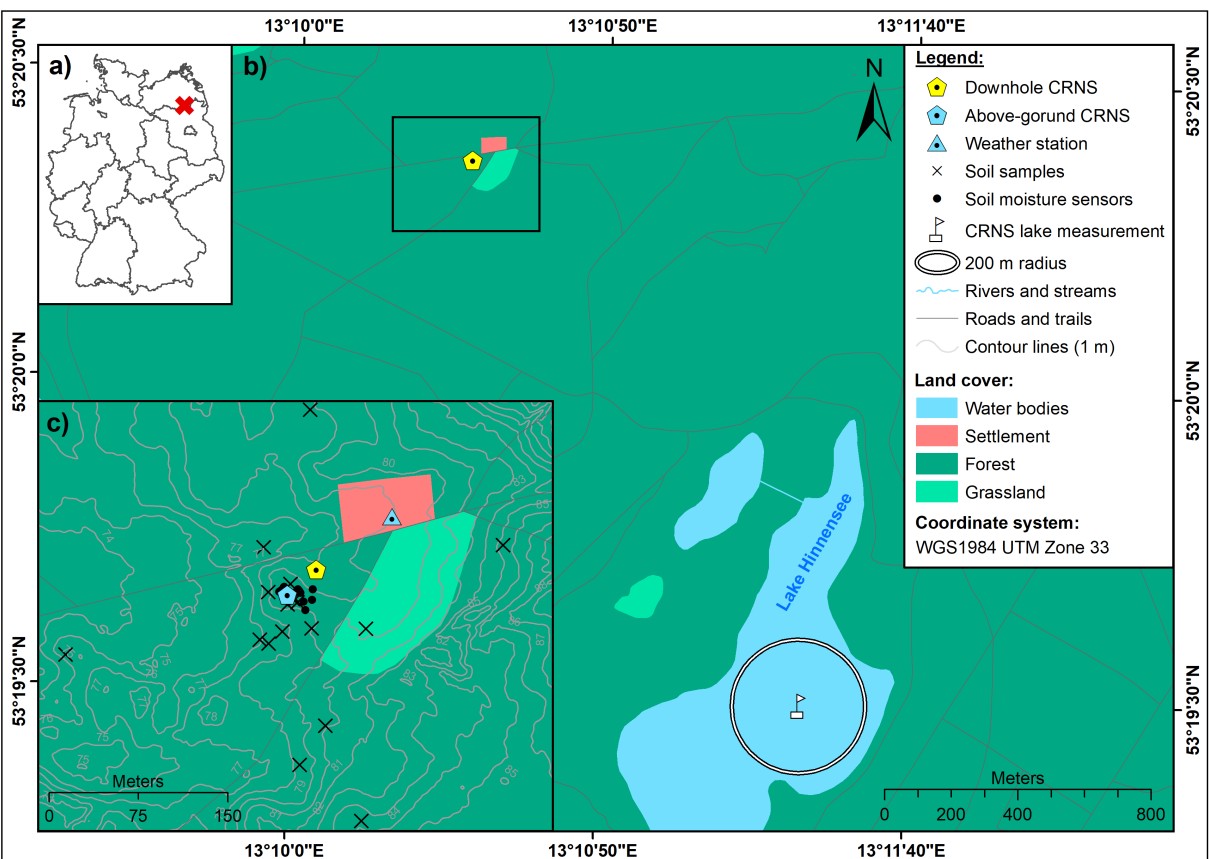

**Figure 1.** Location of the study area within Germany (a), location of the study site in relation to the reference lake measurement site (b) and the details of the CRNS observation site „Serrahn" where the field experiment took place (c) (digital elevation model: LAIV-MV - State Agency for Interior Administration Mecklenburg-Western Pomerania (2011), land cover: BKG - German Federal Agency for Cartography and Geodesy (2018)).



The study site is located on a glacial terminal moraine formed during the Pommeranian phase of the Weichselian glaciation in the Pleistocene (Börner, 2015). The sedimentological profile obtained during the drilling of an on-site groundwater observation well (Fig. 2) with a total depth of 24 m revealed an uppermost layer of aeolian sands deposited during the Holocene reaching a depth of 450 cm followed by a 400 cm thick layer of glacial till which can be attributed to the geological unit of the terminal moraine. From a depth of 850 cm a layer of glacio-fluvial coarse sands containing fine gravel components which extents downward until reaching the glacial till deposited during the earlier Frankfurt phase of the Weichselian glaciation. Regular measurements show a variation of the groundwater level between 13 and 14 m below the surface. Soil samples taken in the scope of the calibration of the permanent CRNS sensor „Serrahn" in February 2019 were taken in order to determine the soil physical characteristics such as average grain size distributions, soil organic matter and lattice water from laboratory analyses shown in Tab. 1. Soil organic matter and lattice water contents were obtained from sub-samples of bulk samples from all sample locations per depth using 24 h loss-on-ignition analyses at 550 and 1000°C, respectively. Based on the average bulk density in 0–35 cm in the upper soil layer and 35 cm depth as the representative value for greater depths, soil water contents at field capacity and wilting point where derived for medium fine sand from tabulated values (Sponagel et al., 2005). Accordingly, 0.16 and 0.06 $cm^3\,cm^{-3}$ for the upper layer as well as 0.12 and 0.04 $cm^3\,cm^{-3}$ for greater depths were derived for the soil water content at field capacity and wilting point. Similarly, the soil porosity was estimated based on the material density of quartz (2.65 $g\,cm^{-3}$) and corrected for the amount of soil organic matter based on the density of cellulose (1.5 $g\,cm^{-3}$). Consequently, we derived a porosity of 0.52 and 0.38 $cm^3\,cm^{-3}$ for the upper soil layer and below depths of 35 cm.

**Table 1.** Soil physical characteristics obtained from laboratory analyses of soil samples taken in February 2019. Soil bulk densities per depth were obtained from oven-drying soil core samples at 105°C for 12 h and subsequent averaging.

| Depth [cm] | Grain size [weight-%] | | | | | Bulk density [$g\,cm^{-3}$] | Organic matter [$g\,g^{-1}$] | Lattice water [$g\,g^{-1}$] |
|---|---|---|---|---|---|---|---|---|
| | > 2 mm | 2 - 0.63 mm | 0.63 - 0.2 mm | 0.2 - 0.063 mm | < 0.063 mm | | | |
| 0–5 | 2.7 | 19.7 | 42.2 | 33.7 | 2.1 | 0.24 | 0.32 | 0.003 |
| 5–10 | 1.1 | 8.7 | 43.5 | 45.7 | 2.4 | 0.77 | 0.10 | 0.002 |
| 10–15 | 0.7 | 7.2 | 41.5 | 47.9 | 2.8 | 1.25 | 0.05 | 0.002 |
| 15–20 | 1.2 | 7.8 | 38.7 | 44.3 | 2.2 | 1.43 | 0.02 | 0.002 |
| 20–25 | 1.7 | 7.7 | 42.2 | 46.5 | 2.2 | 1.55 | 0.02 | 0.002 |
| 25–30 | 1.7 | 8.5 | 43.5 | 45.4 | 1.2 | 1.59 | 0.01 | 0.002 |
| 30–35 | 1.1 | 8.0 | 42.8 | 46.8 | 1.5 | 1.63 | 0.01 | 0.002 |

The landcover at the site is mainly a mixed forest dominated by European beech (*Fagus sylvatica*) and Scots pine (*Pinus sylvestris*) with a clearing covered by grassy vegetation in a few decameters distance. A vegetation survey was conducted in July 2021 in order estimate the total above-ground biomass on-site. Using allometric regressions for *Pinus sylvestris* (Urban et al., 2014) and *Fagus sylvatica* (Chakraborty et al., 2016) revealed a total wet above-ground biomass estimate of 3.73 $g\,cm^{-2}$, assuming that other sources of biomass can be neglected.

Along with the stationary CRNS instruments and the groundwater observation well, the study site is equipped with a weather station and a network of in-situ point-scale soil moisture sensor profiles (type SMT100; Truebner GmbH, Germany). The soil



moisture sensors are installed in depths down to 450 cm along the profiles displayed in Tab. A1 continuously monitoring the volumetric soil moisture based on the manufacturer's calibration function. The measurement interval is 10 minutes. The soil moisture profiles are located close to the CRNS instruments and in a distance of 20 to 40 m to the groundwater observation

well (Fig. 1).

## 2.2 Experimental design

In the scope of this study, we deployed a gaseous proportional neutron detector of the type CRS1000 (Hydroinnova LCC, USA) inside the on-site groundwater observation well. The detector uses $^3$He as the converter gas (see Zreda et al., 2012; Schrön et al., 2018, for details). We disassembled the original setup and placed two unshielded counter tubes into 50 cm long polypropylene

pipes with a wall thickness of 1,9 mm. The relative air humidity in closed groundwater observation wells is constantly close to saturation, making such additional protection of the counter tubes necessary. For the downhole measurements, the CRS1000 counter tubes as well as their readout electronics were lowered into the well to the desired measurement depths by steel ropes. The data logger with its DC power supply remains above the surface.

As shown in the schematic illustration in Fig. 2 the groundwater well itself is composed of an aluminium tube above

the surface mounted into a small concrete foundation while the below-ground tube is made of 7.5 mm thick PVC (polyvinyl chloride) with an inner diameter of 11 mm. An approximately 100 mm wide gap between the surrounding undisturbed sediment and the well tube was filled with sand and clay (see Fig. 2) depending on the surrounding material during the installation of the groundwater well in 2014. The presence of filling material as well as the PVC tube material may reduce the response of the sensor to changes in soil moisture of the surrounding undisturbed soil due to for example the high absorption cross section

of chlorine and scattering cross section of hydrogen contained in the PVC material. Although some influence on the neutron signal has been described for the active neutron probe (e.g. Keller et al., 1990), the precise influence of both remains unknown for the d-CRNS approach. Following the assumption that the soil moisture dynamics in the porous filling material is similar to the surrounding undisturbed material and the sphere of influence largely exceeds the volume of the filling material, we expect the soil moisture signal to dominate the dynamics in the downhole neutron intensity.

Above-ground CRNS relies on epithermal neutrons counted by moderated detector tubes shielded with a 2.5 cm layer of high-density polyethylene (HDPE). In the scope of d-CRNS, we use thermal neutrons counted from unshielded detectors. This is done for different reasons. Firstly, thermal neutrons respond to changes in environmental hydrogen content and thus, soil moisture (e.g., Hubert et al., 2016; Weimar et al., 2020; Rasche et al., 2021). Secondly, we expect the neutron intensity (i.e. count rate) to decrease strongly with soil depth. A bare counter tube is then more effective as the HDPE shielding of a

moderated tube would not only slow down but also reflect a certain percentage of potentially countable neutrons away from the instrument and would thus reduce the observed intensity. Furthermore, it has been shown that thermal neutrons can be potentially used to obtain soil moisture information from larger depths compared to epithermal neutrons (Rasche et al., 2021). This may be especially useful for downhole measurements in order to increase the potential measurement radius. Lastly, using unshielded detector tubes is of practical nature as the removal of the 2,5 cm HDPE shielding reduces the weight and dimensions

of the CRS1000 counter tubes allowing them to fit into standard groundwater well tubes with an inner diameter of 11 cm.





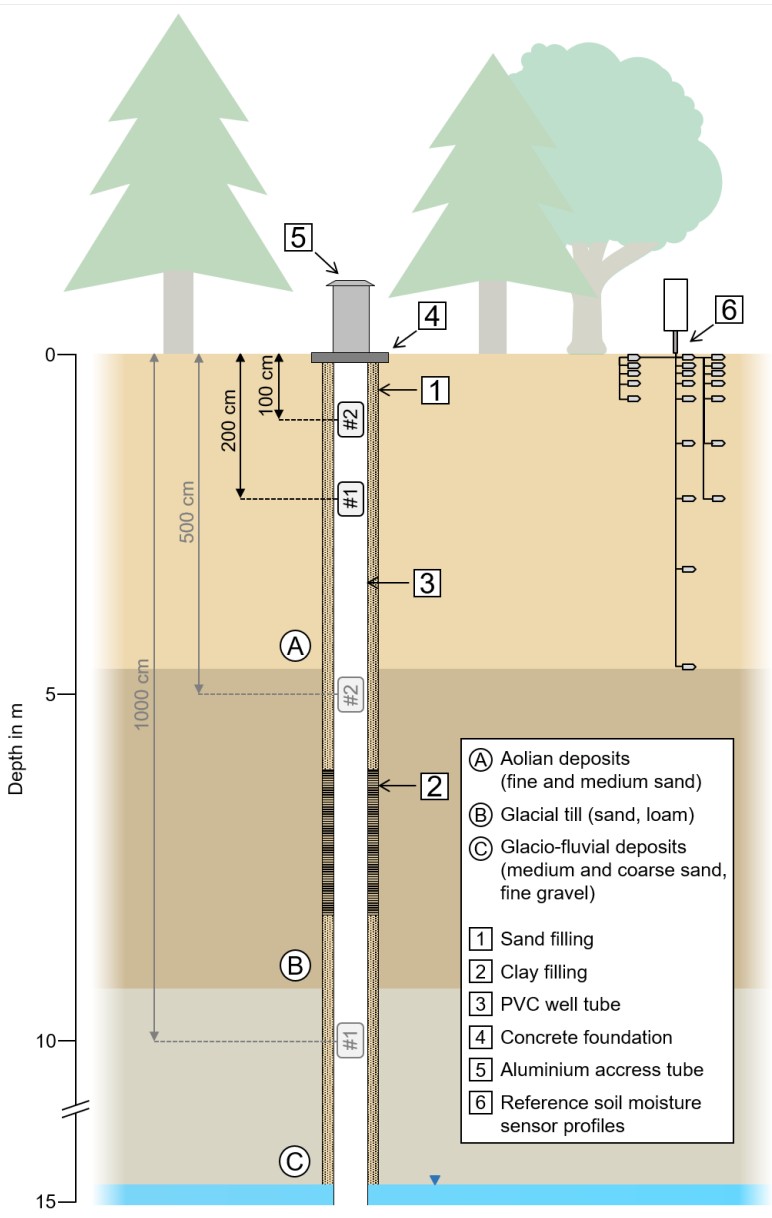

**Figure 2.** Schematic illustration of the experimental set-up at the study site. The thermal (unshielded) neutron detectors #1 and #2 were simultaneously installed in 100 and 200 cm from July to November 2021 and January to May 2022 and in 500 and 1000 cm for the time period in between.





### 2.3 Neutron measurements and processing

The counter tubes simultaneously measured neutron intensities in 100 (tube no. 2) and 200 cm depth (tube no. 1) from July 2021 to November 2021, followed by a second period from January 2022 to May 2022. In between these two periods, the detectors were placed in 500 (tube no. 2) and 1000 cm depth (tube no. 1).

Following conventional approaches, the observed above-ground epithermal neutron intensities would be corrected for variations in atmospheric shielding depth, absolute air humidity and primary neutron flux (Zreda et al., 2012; Rosolem et al., 2013) before being smoothed by a 25 h and 49 h moving average in order to reduce the uncertainty in the time series (Schrön et al., 2018). For downhole CRNS applications measuring thermal neutrons an adjusted correction algorithm for the neutron signal is required. Thermal neutrons detected by a downhole CRNS have not interacted with the atmosphere from the point where
they reach water-sensitive energies in the soil to eventually reaching the detector. As a consequence, downhole thermal neutron intensities will be corrected for variations in atmospheric shielding depth and primary neutron influx only. In accordance with Heidbüchel et al. (2016), the required neutron attenuation length was set to $135.9\,\mathrm{g\,cm^{-2}}$ for the study site. The neutron monitor database (www.nmdb.eu, station: JUNG - Jungfraujoch) is used to obtain data for the primary neutron flux.

    For converting above-ground neutron intensities to soil moisture estimates a calibration against soil moisture reference
measurements is necessary in order to scale the transfer function to possible site-specific characteristics. This is the case for the N0 method (Desilets et al., 2010), as well as the recently introduced universal transport solution (UTS) (Köhli et al., 2021). Reference soil moisture measurements can be obtained in shallow depths from soil sampling and subsequent laboratory analysis or from point-scale in-situ soil moisture sensors. However, using a CRNS detector as a downhole instrument would require reference measurements from greater depths which are more difficult to acquire. For this reason, we decided to adapt
an approach proposed by Franz et al. (2013) for above-ground CRNS applications using epithermal neutrons. It comprises scaling the transfer function by the neutron intensity measured above water instead of above dry soil as it is the case for the N0 method. The count rate above dry soil can be calibrated with reference soil moisture information but without this reference information the count rate above dry soil needs to be measured in order to transfer observed neutron count rates to soil moisture. Measuring neutrons above an (ideal) hydrogen-free soil is practically impossible while the intensity above
water can be measured directly and thus, does not require additional calibration. This approach has been applied in previous studies related to above-ground epithermal CRNS (e.g., McJannet et al., 2014; Andreasen et al., 2016, 2020). We adjusted the approach of above-water measurements for the scaling of thermal neutrons. Therefore, a measurement at 25 cm (detector bottom) above the water surface and of 1.5 h duration was conducted with the two detector tubes on Lake Hinnensee prior to their installation below ground (Fig. 1). Scaling the downhole neutron intensities with the detector-specific neutron intensity
above water by calculating the neutron ratios Eq. (1) allows for comparison of observed neutron ratios with simulated neutron ratios. It furthermore enables the development of a transfer function from simulations which may be applied without additional calibration against reference soil moisture measurements.

    Unlike the measurements below ground, the neutron intensity above water needs to be corrected for variations in absolute air humidity. A specific humidity correction function has been developed for epithermal neutrons only (Rosolem et al., 2013;





Köhli et al., 2021) and may not be valid for thermal neutrons. For this reason, we developed a first equation to correct thermal neutron intensities observed above water to variations in absolute humidity. The observed neutron ratio $N_r$

$$N_r = \frac{N_s}{N_w} \qquad (1)$$

can then be calculated from the downhole thermal neutron intensity $N_s$ corrected for variations in atmospheric shielding depth and primary neutron influx only as well as the thermal neutron intensity above water $N_w$ corrected for variations in atmospheric shielding depth, primary neutron influx and absolute humidity.

## 2.4 Particle transport simulations

Several different Monte Carlo-based particle transport simulation toolkits have previously been used for the investigation of secondary cosmic-ray neutrons at the soil-atmosphere interface in the context of CRNS including GEANT4 (Hubert et al., 2016; Brall et al., 2021), MCNP (Zreda et al., 2008; Franz et al., 2012; Andreasen et al., 2016, 2017; Weimar et al., 2020; Köhli et al., 2021) and URANOS (Köhli et al., 2015, 2021; Li et al., 2019; Rasche et al., 2021) which only simulates neutrons of different energies. Although simulating only neutrons and not e.g. protons and muons as well might be sufficient at the soil-atmosphere interface with a detector placed above the surface, the simulation of the neutron flux in different depths of the soil requires the inclusion of several other types of particles that induce neutron production in the soil volume itself. That is because the atmospheric neutron flux is attenuated strongly within the soil volume and in-soil neutron production dominates the thermal neutron flux below several decimetres soil depth. Consequently, we used the model MCNP v6.2 (Werner et al., 2018) to simulate the neutron ratios for different soil depths and soil bulk densities as this model includes e.g. protons, muons and neutrons as source particles within the model domain. Energy spectra and angular distributions of the particle species were set according to Sato (2015) and Sato (2016). The starting particles are released 450 m above the soil surface embedded in a cylindrical simulation domain with 6 m radius and reflecting boundaries.

All simulation scenarios described in the following comprise a cutoff rigidity of 2.6 GeV, an absolute humidity of $10 \, \text{g} \, \text{m}^{-3}$ and an atmospheric pressure of 1013.25 hPa. The detector has a length of 50 cm and diameter of 5.5 cm. The tube volume is filled with $^3$He at 1.5 bar and all neutrons are counted that undergo an absorption process in the simulated detector volume. Thus, the behaviour of an unshielded (bare) proportional neutron detector tube is modelled.

In a first step, the detector was placed 50 cm above an infinite water surface, with the 50 cm being measured from the detector tube centre. The detector volume is slightly larger than the real CRS1000 detector tubes in order to enhance the counting statistics in low count environments. To estimate the influence of variations in absolute humidity on the thermal neutron intensity above water and in order to develop a correction function, the simulation scenario was repeated with air humidity values of 1, 6, 11, 16, 21 and $26 \, \text{g} \, \text{m}^{-3}$.

Neutron responses in different soil depths were modelled with a soil bulk density of $1.43 \, \text{g} \, \text{cm}^{-3}$ where the soil material is composed of 75 % $SiO_2$ and 25 % $Al_2O_3$. The detector was placed in shielding depths of 75, 100, 150, 200, 250, 300, 350, 400, 500, 750, 1000 and $1500 \, \text{g} \, \text{cm}^{-2}$ with ten different soil moisture contents ranging from 0.005 to $0.5 \, \text{cm}^3 \, \text{cm}^{-3}$. The shielding





depth describes the total amount of matter a particle has to travel trough. It is influenced by the material dry bulk density, the absolute depth in cm and the soil water content (Eq. (9)). In the simulation scenarios, the absolute depth of the detector was changed for the different soil moisture states in order to maintain the same simulated shielding depth at the detector centre.

In accordance with the setup of the real groundwater observation well, the virtual detector was placed in a PVC cylinder of the same dimensions. To investigate the influence of the local soil bulk density on the simulated neutron response, a smaller subset of scenarios for all soil moisture states with the shielding depths of 75, 100, 200, 350, and 500 $\mathrm{g\,cm^{-2}}$ were additionally modelled with soil bulk densities of 1.1 and 1.8 $\mathrm{g\,cm^{-3}}$.

In order to assess the sphere of influence for the downhole neutron detector, particle tracking simulations for a single shield-
ing depth of 300 $\mathrm{g\,cm^{-2}}$ and all soil moisture conditions listed above were run. The trajectory of all detected neutrons is traced backwards in order to determine the locations where they probed the soil via scattering. Above 150–200 $\mathrm{g\,cm^{-2}}$ an increase of the measurement volume can be expected caused by an increased contribution from neutrons which previously scattered in the atmospheric layer before entering the soil and eventually being detected. Using the simulations with 300 $\mathrm{g\,cm^{-2}}$ allows for the isolated characterisation of the measurement volume without the influence of neutrons which previously interacted with
the atmosphere. In the scope of this study, the dimensions of the measurement volume are estimated based on the locations of all elastic scattering processes above the thermal energy regime and thus the entire moderation process from the point where a detected neutron was generated.

To assess the influence of the well tube material on the neutron ratio as well as on the dimensions of the sphere of influence, we not only simulated a PVC well tube with a wall thickness of 7.5 mm, but also a well tube composed of stainless steel
of equal wall thickness as well as thinner PVC. The additionally simulated PVC material had wall thickness of 5 mm and a density of 1.44 $\mathrm{g\,cm^{-3}}$ while the steel tube (type X5CrNi18-10) had a wall thickness of 7.5 mm and a density of 7.85 $\mathrm{g\,cm^{-3}}$ and contained 18 % chromium as well as 10 % nickel. Particle transport simulations with a shielding depth of 300 $\mathrm{g\,cm^{-2}}$ were run for investigating the size of the measurement volume for stainless steel and the neutron ratios were simulated for a soil bulk density of 1.43 $\mathrm{g\,cm^{-3}}$ and shielding depths from 100 to 400 $\mathrm{g\,cm^{-2}}$.

## 3 Results

### 3.1 MCNP simulations

#### 3.1.1 Neutron ratio response and sphere of influence

We simulated the detector-specific neutron intensity above water to aid in processing the downhole neutron intensities without the need for calibration based on in-situ soil moisture information. The simulation revealed a dependence of thermal neutrons
detected above water on absolute air humidity. The thermal neutron intensity decreased approximately linearly by 0.21 % per 1 $\mathrm{g\,m^{-3}}$ absolute humidity ($R^2 = 0.93$) which is less than half of what has been reported for epithermal neutrons (Rosolem et al., 2013). The correction function developed by Rosolem et al. (2013) for epithermal neutrons can thus be adjusted to correcting observed thermal neutron intensities above water by changing the correction factor from the original 0.0054 to the





derived 0.0021. It should be noted that the reference absolute humidity for the simulations and the transfer functions was set to
an arbitrary $10\,\mathrm{g\,cm^{-3}}$.

Neutron ratios were calculated for all neutron transport simulations using the reference simulation scenario with the detector placed above a water surface. The simulation results for the first set of scenarios conducted with a soil bulk density of $1.43\,\mathrm{g\,cm^{-3}}$ are shown in Fig. 3. The response of the simulated neutron ratios observed by the virtual downhole neutron detector to changes in soil moisture differ between the different simulated shielding depths with generally lower neutron ra-
tios in larger depths. For each simulated shielding depth, the neutron ratio decreases with increasing soil moisture although a specific behaviour can be observed for shallow shielding depths. From the 75 to the $100\,\mathrm{g\,cm^{-2}}$ shielding depth scenario the simulated neutron ratio increases, i.e. the neutron intensity observed by the downhole neutron detector increases, when the soil moisture content is below $0.045\,\mathrm{cm^3\,cm^{-3}}$. This reveals a peak neutron intensity in shallow soil layers under low soil moisture conditions. At higher soil moisture contents, this peak ratio disappears and a continuous decrease of the neutron
ratio with increasing shielding depth per simulated soil moisture content can be observed. The simulation sets conducted with lower ($1.1\,\mathrm{g\,cm^{-3}}$) and higher ($1.8\,\mathrm{g\,cm^{-3}}$) soil bulk densities show a similar behaviour, although the absolute values of the neutron ratios change. Higher soil bulk densities result in lower neutron intensities and thus, lower neutron ratios observed by the virtual downhole neutron detector and vice versa (see section 3.1.2 for details). We also investigated possible influences of the groundwater observation well tube material by simulating a 5 mm PVC and 7.5 mm stainless steel tubing. The additional
subset of simulations revealed that neutron ratios for a well tube composed of PVC are on average 60 % lower compared to a stainless steel tube with equal wall thickness but respond similarly to changes in soil moisture. In addition, a PVC material with a thickness of 7.5 mm produces neutron ratios which are on average 28 % lower compared to a thinner PVC tubing with a wall thickness of only 5 mm.

In this study, the sphere of influence, i.e. the measurement volume around the downhole neutron detector, is calculated as the
86 %-quantile and 95 %-quantile of all locations of elastic collision processes of a detected neutron in the soil. The definition based on the 86 %-quantile relies on the convention established for describing the horizontal integration radius and integration depth of above-ground CRNS applications while the 95 %-quantile is common for active neutron probe applications. In order to better compare d-CRNS with above-ground CRNS and the traditional active neutron probe, the results of both definitions are shown in Fig. 4.

In line with above-ground CRNS, only the the 86 %-quantile is used for a mathematical description of the sphere of influence and its dimensions. The shape of the sphere of influence simulated for a neutron detector with a height of 50 cm and diameter of 5.5 cm and for different soil moisture contents in a shielding depth of $300\,\mathrm{g\,cm^{-2}}$ measured at the detector centre can be described by the following equations. The horizontal sensitive radius of the sphere $R_{86}$ can be described by the local soil moisture content $\theta$ in $\mathrm{cm^3\,cm^{-3}}$ (see Eq. (8)) as well as the local soil bulk density $\rho$ ($\mathrm{g\,cm^{-3}}$) and the fitted parameters $p_1$ to
$p_3$ (Tab. 2) at the depth of the neutron detector via

$$R_{86} = \frac{p_1}{(\rho/(\mathrm{g\,cm^{-3}})) \cdot (1 + p_2 \cdot \theta \cdot 100)} + p_3 \cdot \left(\rho/\left(\mathrm{g\,cm^{-3}}\right)\right)^{p_4}. \tag{2}$$





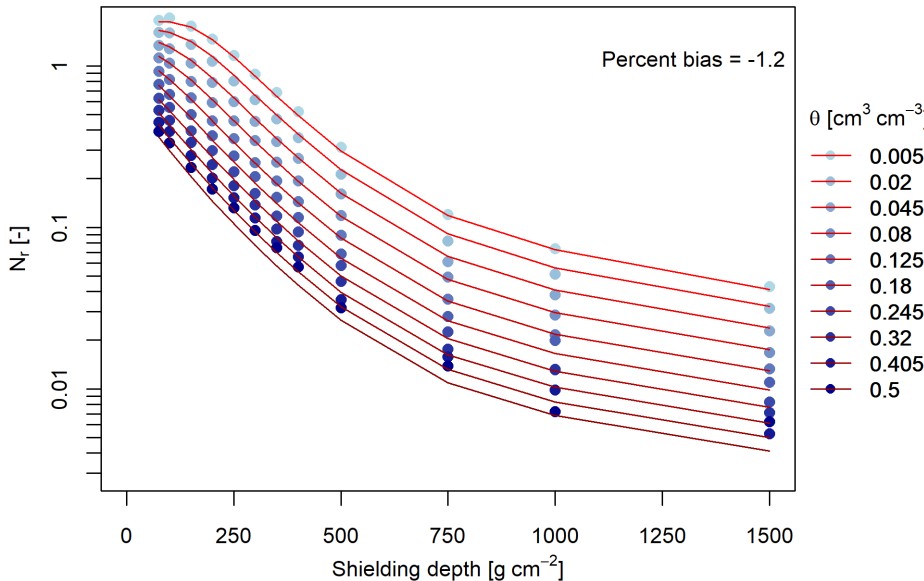

**Figure 3.** Simulated values of $N_r$ from neutron transport modelling with the predicted values (red lines) from Eq. (5-9) for a soil bulk density of 1.43 g cm$^{-3}$, different soil moisture conditions and shielding depths.

The simulated horizontal radii of the sphere of influence defined as the $R_{86}$ and $R_{95}$ are shown in Fig. 4 for different soil moisture values and soil bulk densities. Both, $R_{86}$ and $R_{95}$ decrease with increasing soil moisture and show generally lower values when the soil bulk density is higher. For instance, a bulk density of 1.43 g cm$^{-3}$ leads to a $R_{95}$ of 170 cm at 0.005 cm$^3$ cm$^{-3}$

and to 34 cm at 0.5 cm$^3$ cm$^{-3}$. At the same bulk density and for the same soil moisture values, $R_{86}$ is generally smaller and decreases from 127 to 26 cm. The values derived from Ølgaard's (1965) equation in Gardner (1986) for the $R_{95}$ of an active neutron probe reveal 69 and 16 cm at 0.005 and 0.5 cm$^3$ cm$^{-3}$ and hence, smaller radii of the sphere of influence than for a downhole neutron detector for all simulated soil bulk densities and both, $R_{95}$ and $R_{86}$. Consequently, even for a high bulk density of 1.8 g cm$^{-3}$ the generally smaller $R_{86}$ of d-CRNS is approximately 40 % larger than the $R_{95}$ of an active neutron

probe.

The average vertical sensitive radius of the sphere of influence $V_{86}$ can be described by Eq. (3) and has a size range of 89 to 24 cm from the lowest (0.005 cm$^3$ cm$^{-3}$) to the highest (0.5 cm$^3$ cm$^{-3}$) simulated soil moisture content at 1.43 g cm$^{-3}$ bulk density. In combination with the simulated horizontal radii, an ellipsoidal shape of the sphere of influence can be derived for a downhole neutron detector in d-CRNS applications with $R_{86}$ and $V_{86}$ describing the ellipsoids semi-axes from the detector

centre. However, a vertical shift of the most sensitive area, i.e. the location of the largest horizontal radius relative to the detector centre with varying soil moisture, can be observed. This dimension is described by Eq. (4) and shown in Fig. 5. The vertical sensitive radius $V_{86}$ increases with decreasing soil moisture while the most sensitive region $Vc_{86}$ is always located



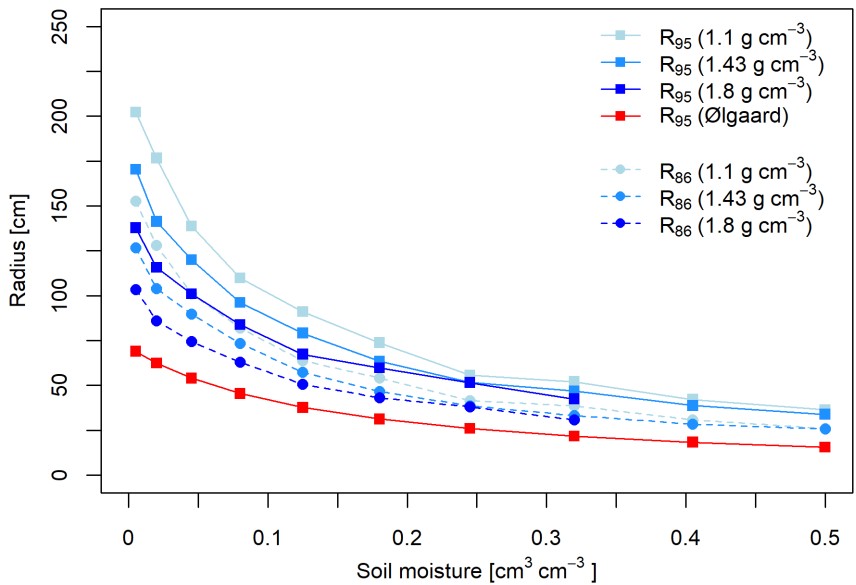

**Figure 4.** Simulated horizontal radii of the sphere of influence defined as the $R_{86}$ and $R_{95}$ for different local (in the depth of measurement) soil moisture values $\theta$ (see Eq. (8)) and soil bulk densities. In addition, the $R_{95}$ values based on Ølgaard's (1965) equation in Gardner (1986) for an active neutron probe are displayed for comparison.

slightly above the detector centre and shifts upwards with lower soil water contents. For example, for a soil bulk density of 1.43 g cm$^{-3}$, the most sensitive region of the downhole neutron detector $Vc_{86}$ is located 5 to 20 cm (for 0.5 to 0.005 cm$^3$ cm$^{-3}$ soil moisture content) above the detector centre. The fitted parameters $p_1$ to $p_5$ required in Eq. (2), Eq. (3) and Eq. (4) can be found in Tab. 2 and a schematic illustration of the sphere of influence for different soil moisture contents can be found in Fig. 6. We define the vertical footprint size as:

$$V_{86} = \frac{p_1}{(\rho/(\mathrm{g\,cm}^{-3}))^{p_2} \cdot (1 + p_3 \cdot \theta \cdot 100)} + p_4 \cdot \left(\rho/(\mathrm{g\,cm}^{-3})\right)^{p_5}, \tag{3}$$

and the location of the most sensitive region above the detector center as:

$$Vc_{86} = \frac{p_1 \cdot \exp\left((-\theta \cdot 100)/p_2\right)}{(\rho/(\mathrm{g\,cm}^{-3}))} + p_3 \cdot \left(\rho/(\mathrm{g\,cm}^{-3})\right)^{p_4}. \tag{4}$$

The simulation results for a well tube made from stainless steel with a wall thickness of 7.5 mm revealed similar dimensions of the sphere of influence compared to those derived for PVC with equal wall thickness. Averaged over the range of simulated soil moisture values and for a soil bulk density of 1.43 g cm$^{-3}$, $R_{86}$ is approximately 1.2 cm larger for a well tube made from



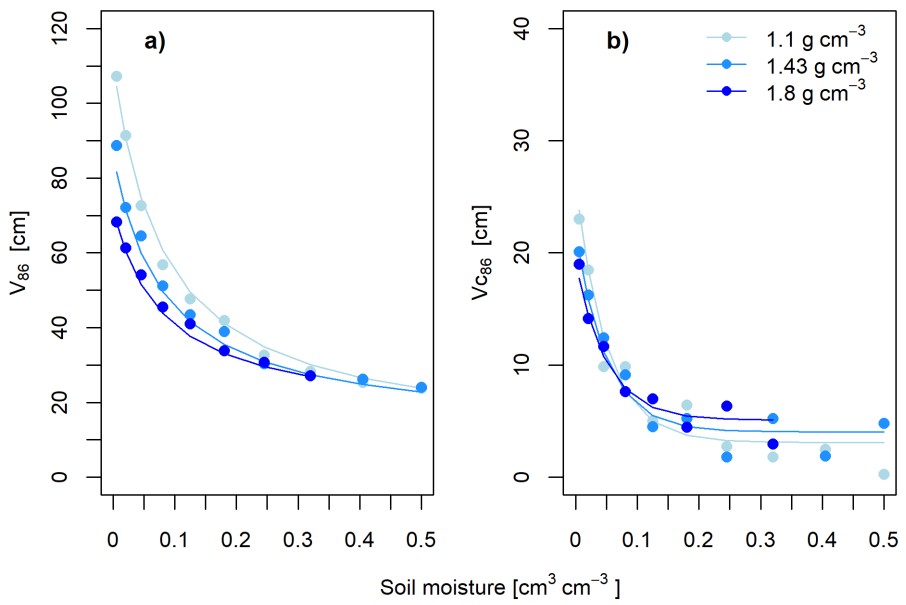

**Figure 5.** (a) Simulated values of the vertical sensitive radius $V_{86}$ from the detector centre and (b) the position of the most sensitive area relative to the detector centre $Vc_{86}$ for different local soil moisture values $\theta$ (see Eq. (8)) and soil bulk densities.

PVC compared to the steel tube while $Vc_{86}$ and $V_{86}$ are 8.2 cm and 4.5 cm smaller. The fitted parameters for a well tubing made
from stainless steel can be found in Tab. A2.

### 3.1.2   Predicting neutron ratios

Our neutron transport simulations revealed a change of the hyperbolic relationship between the neutron ratio $N_r$ and the simulated soil moisture content depending on the shielding depth $D$ measured at the centre of the detector tube (Fig. 3). Therefore, we derived a hyperbolic fit model with the analytical form of Eq. (5) for each shielding depth and subsequently predicted the
shape-defining parameters $F_1$ and $F_2$ by shielding depth. A third-order and second-order exponential model resulted in a high goodness-of-fit for parameters $F_1$ and $F_2$, which lead to the following equations allowing for the estimation of $N_r$:

$$N_r = \frac{F_1}{F_2 + \theta}, \tag{5}$$

where    $F_1 = (p_1 \cdot \exp(p_2 \cdot D) + p_3 \cdot \exp(p_4 \cdot D) + p_5 \cdot \exp(p_6 \cdot D)) \cdot \dfrac{\rho}{1.43\,\mathrm{g\,cm^{-3}}}, \tag{6}$

$$F_2 = (p_7 \cdot \exp(p_8 \cdot D) + p_9 \cdot \exp(p_{10} \cdot D)) \cdot \frac{\rho}{1.43\,\mathrm{g\,cm^{-3}}}. \tag{7}$$



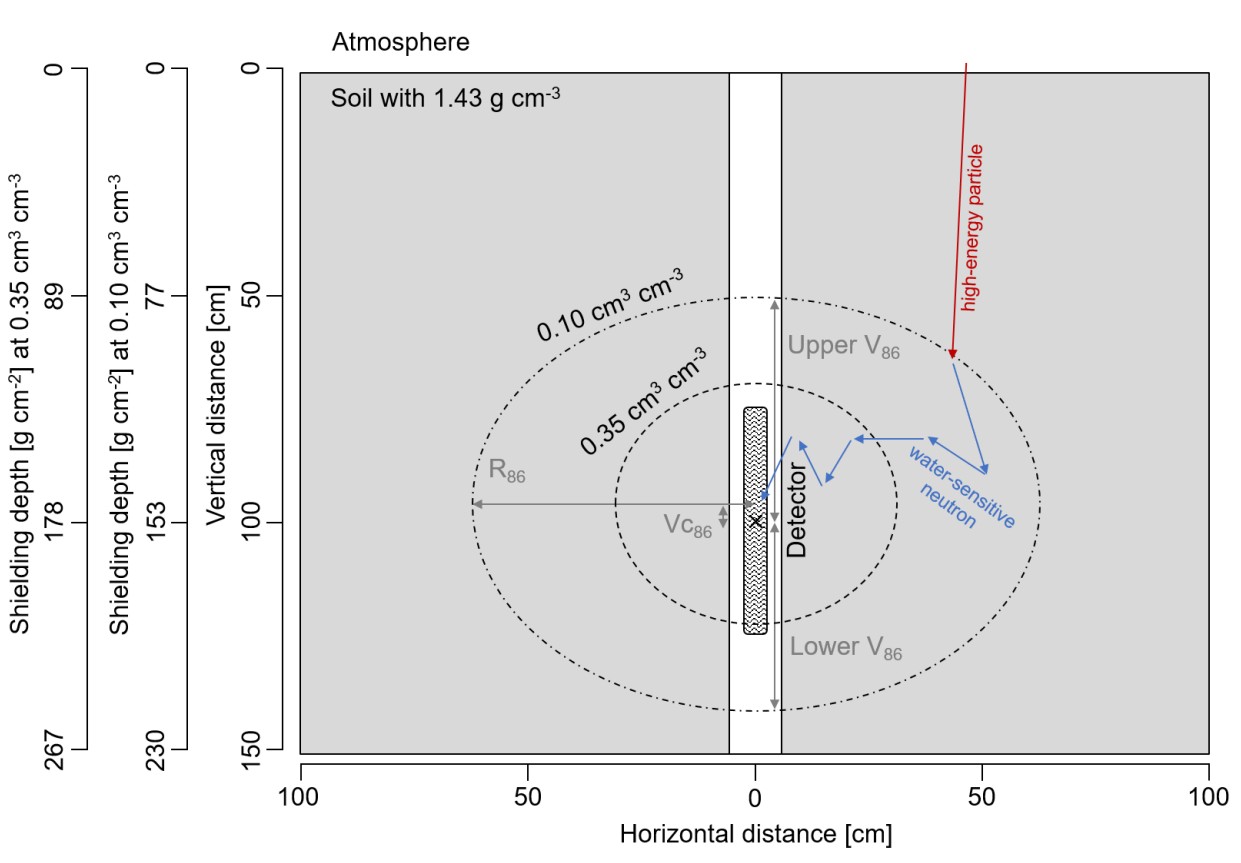

**Figure 6.** Schematic illustration of the sphere of influence described by $R_{86}$, $Vc_{86}$ and $V_{86}$ for a soil with a bulk density of $1.43\,\mathrm{g\,cm^{-3}}$ and a soil moisture content of 0.10 and $0.35\,\mathrm{cm^3\,cm^{-3}}$ in the entire soil column. The $V_{86}$ is different above (upper $V_{86}$) and below (lower $V_{86}$) the neutron detector (its center here marked by an X) and thus, $V_{86}$ represents the average vertical extent. A schematic neutron transport path is displayed with a high-energy particle producing a hydrogen-sensitive and thus, water-sensitive neutron in the soil which is slowed down to thermal energies by multiple elastic scattering interactions before eventually being detected.

The equation makes use of two key quantities, the local soil water content in the depth of measurement

$$\theta = \theta_{\mathrm{SM}} + \theta_{\mathrm{SOM}} + \theta_{\mathrm{LW}} \qquad (8)$$

and the shielding depth

$$D = d \cdot \left( \hat{\rho} + \hat{\theta}_{\mathrm{SM}} \cdot \rho_{\mathrm{water}} \right) + D_{\mathrm{AGB}}. \qquad (9)$$

The parameters and variables used in Equations (5) to (9) are explained in the following. The fitted parameters $p_1$ to $p_{10}$ can
be found in Tab. 2 for a PVC well tube with wall thickness of 7.5 mm, in Tab. A2 for a well tube composed of stainless steel of equal thickness and in Tab. A3 for a PVC well tube with a thickness of only 5 mm while the following remaining variables





depend on the conditions of the study site. The variable $\theta$ describes the total local water content comprising soil moisture $\theta_{SM}$, lattice water $\theta_{LW}$ and the water equivalent of soil organic matter $\theta_{SOM}$ in $cm^3\,cm^{-3}$ in the depth of neutron observation measured as the distance from the soil surface to the centre of the detector tube $d$ in cm. Based on the subset of neutron transport

simulations with different soil bulk densities, we found $F_1$ and $F_2$ to be dependent on the ratio between the density $\rho$ in the depth of measurement and the soil density of $1.43\,g\,cm^{-3}$ used in the simulations from which the equations were derived.

The second required variable is the shielding depth $D$ describing the total mass neutrons, protons and muons need to travel through before reaching the depth of the centre of the detector tube. The shielding depth represents the integral mass from the surface to the detector centre. Thus, it depends on the measurement depth $d$ in cm as well as on the average soil bulk density $\hat{\rho}$

$(g\,cm^{-3})$ from the soil surface to the measurement depth. Likewise, the average soil water content $\hat{\theta}_{SM}$ $(cm^3\,cm^{-3})$ is required, assuming a constant water density $\rho_{water}$ of $1\,g\,cm^{-3}$. It should be noted that for the calculation of the shielding depth, the total mass of material above the centre of the detector tube is required, regardless of its elemental composition. The mass of soil organic matter and lattice water is thus already accounted for by the integral soil bulk density. Additionally, as the study site of the present study is located in a mixed forest, the total above-ground mass $(D_{AGB})$ associated with vegetation needs

to be added. Further, it should be noted that in Eq. (5–7) the parameters $p_2$, $p_4$, $p_6$, $p_8$ and $p_{10}$ have the unit $cm^2\,g^{-1}$ (inverse shielding depth) while the remaining parameters $p_1$, $p_3$, $p_5$, $p_7$ and $p_9$ are dimensionless.

Applying the above equations (Eq. (5-9)) to the input variables of the neutron transport simulation and comparing this prediction with the simulated $N_r$ shows a good overall fit, with a percent bias between the predicted and simulated values of -1.2 % for a bulk density of $1.43\,g\,cm^{-3}$ (Fig. 3) and 0.8 % for all modelled densities.

**Table 2.** Fitted parameters for Eq. (2-7) derived from particle transport simulation scenarios for a PVC well tube with a wall thickness of 7.5 mm.

| Eq. no. | Variable | $p_1$ | $p_2$ | $p_3$ | $p_4$ | $p_5$ | $p_6$ | $p_7$ | $p_8$ | $p_9$ | $p_{10}$ |
|---------|----------|-------|-------|-------|-------|-------|-------|-------|-------|-------|----------|
| (2) | $R_{86}$ | 173 cm | 0.214 | 4.05 cm | 2 | | | | | | |
| (3) | $V_{86}$ | 113 cm | 1.2 | 0.121 | 8.61 cm | 1 | | | | | |
| (4) | $Vc_{86}$ | 25.2 cm | 5 | 2.82 cm | 1 | | | | | | |
| (6) | $F_1$ | 0.252 | -0.0206 $cm^2\,g^{-1}$ | 0.00794 | -0.000839 $cm^2\,g^{-1}$ | 0.267 | -0.00674 $cm^2\,g^{-1}$ | | | | |
| (7) | $F_2$ | | | | | | | 0.0406 | 0.000139 $cm^2\,g^{-1}$ | 0.265 | -0.0172 $cm^2\,g^{-1}$ |

### 3.1.3   Estimating soil moisture

A key motivation of this study is to derive soil moisture time series by d-CRNS in depths larger than those accessible by surface CRNS. The equations above describe the physical relationships that influence the neutron intensity, and thus $N_r$, inside the shaft of the groundwater observation well or access tube. They illustrate that both, the soil moisture in the local depth of the detector as well as the average soil moisture in the vadose zone above the detector have an effect on the $N_r$ observed in a certain

measurement depth. However, estimating two unknown variables, namely $\theta_{SM}$ and $\hat{\theta}_{SM}$, from $N_r$ as the single known variable





is only possible with further assumptions. One option would be the use of Eqs. (5-9) as a forward operator in combination with soil hydraulic models (e.g., HYDRUS-1D, Šimůnek et al., 2008) to model soil moisture time series in different soil depths. The model can then be calibrated by applying Eq. (5-9) with the modelled soil moisture time series and optimising the goodness-of-fit between the observed and predicted $N_r$ by adjusting the parameters in the soil hydraulic model. However, soil hydraulic

models may require further variables such as rainfall, evapotranspiration and root distributions which are not always available.

We propose an alternative, more simple approach to estimate the soil moisture time series in the depth of measurement from the observed neutron ratios by using Eq. (5-9). Following these equations, the approach is based on the fact that the influence of $\theta_{SM}$ on $N_r$ is considerably larger than that of $\hat{\theta}_{SM}$. For example, at a $\theta_{SM}$ of $0.1\,\mathrm{cm^3\,cm^{-3}}$, a change of $\hat{\theta}_{SM}$ from 0.05 to $0.15\,\mathrm{cm^3\,cm^{-3}}$ results in a change of $N_r$ by 6 % (at a measurement depth of 100 cm). In contrast, changing $\theta_{SM}$ from 0.05 to

$0.15\,\mathrm{cm^3\,cm^{-3}}$ at a value of $0.1\,\mathrm{cm^3\,cm^{-3}}$ for $\hat{\theta}_{SM}$ leads to a change of $N_r$ by 47 %. The higher sensitivity of $N_r$ to changes of the soil moisture content in depth of measurement allows for its estimation in the following way:

1. For $\hat{\theta}_{SM}$, we assign values in the range from wilting point to field capacity in steps of $0.001\,\mathrm{cm^3\,cm^{-3}}$.

2. For every value of $\hat{\theta}_{SM}$, we apply Eqs. (5-9),

3. for each time step of the observed time series of $N_r$, values of $N_r$ are calculated by assigning values from $0.01\,\mathrm{cm^3\,cm^{-3}}$

to the soil moisture content at saturation in steps of $0.0001\,\mathrm{cm^3\,cm^{-3}}$ to $\theta_{SM}$.

4. The value of $\theta_{SM}$ that produces the smallest absolute difference between the observed and calculated $N_r$ at each time step is chosen. This procedure results in a time series of $\theta_{SM}$ for each value of $\hat{\theta}_{SM}$.

5. Based on this set of time series, we propose to average the values for $\theta_{SM}$ for each time step in order to provide a single time series of estimated soil moisture values in the depth of measurement. The minimum and maximum time series can

also be calculated to assess the range (uncertainty) of possible $\theta_{SM}$ values based on the observed $N_r$.

## 3.2 Experimental evidence

### 3.2.1 Observed neutron response

The reference measurement at Lake Hinnensee was conducted in order to the derive the detector-specific raw count rate above water and resulted in values of 315 and 155 counts per hour (cph) for detector tube no. 1 and 2, respectively. The measurement

duration of 1.5 h and measurement intervals of one minute led to a Poisson standard deviation of 14 and 10 cph or a coefficient of variation of 4.6 and 6.6 %, respectively. The average uncorrected downhole neutron count rate in 100 cm (tube no. 2) was 101 cph and 70 cph in 200 cm (tube no. 1) covering the entire measurement period. During about two months of measurements at larger depths, the average count rate was significantly lower, with tube no. 2 observing 6 cph in 500 cm and tube no. 1 observing 11 cph in 1000 cm. As a result, the measurement uncertainty increased sharply. For instance, for an observed count

rate of 10 cph the coefficient of variation was 32 %.





$N_r$ calculated based on the corrected neutron count rates decreases with increasing measurement and shielding depth (Fig. 7). The average $N_r$ decreases from 0.63 in 100 cm and 0.22 in 200 cm to 0.039 in 500 cm and 0.034 in 1000 cm.

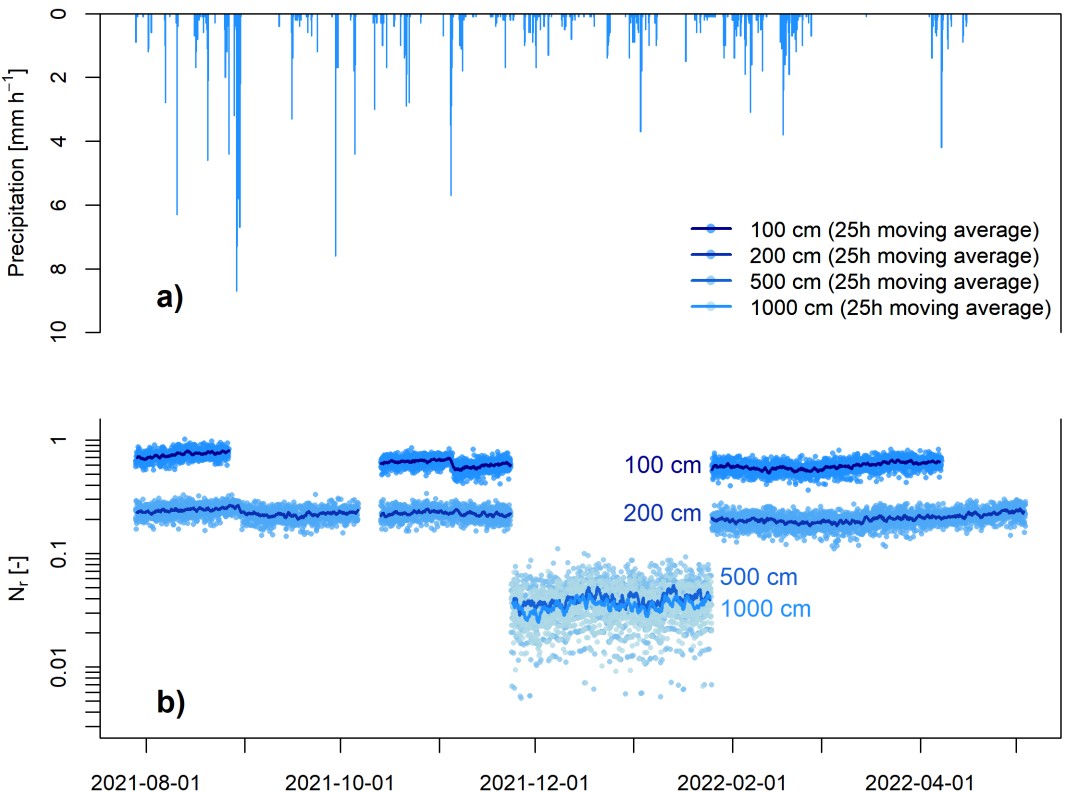

**Figure 7.** (a) Hourly precipitation observed at the study site during the experimental period, (b) observed time series of $N_r$ in the different measurement depths of 100, 200, 500 and 1000 cm. Points represent the original (not smoothed) neutron ratios from corrected neutron intensities while lines represent the 25 h moving average calculated from corrected neutron intensities.

Figure 8 shows the comparison of the observed neutron intensity corrected for variations in air pressure and primary neutron influx $N_s$ and the local in-situ reference soil moisture in 100 and 200 cm depth. Due to the location of all reference sensors outside the expected sphere of influence of the downhole neutron detector, a comparison with the soil moisture sensor in the respective depth showing the highest Pearson correlation coefficient between the observed and predicted values from a hyperbolic non-linear least squares fit model in the form of Eq. (5) is shown. In case of the downhole neutron detector installed at 100 cm depth, the reference sensors with the highest and lowest goodness-of-fit in 70 and 130 cm are considered (due to lack of sensors at 100cm depth), while for the neutron detector at 200 cm depth reference soil moisture sensors in the same depth are available. Figure 8 illustrates that in both measurement depths a distinct neutron intensity response to changes in the local soil moisture following a hyperbolic relationship can be observed. However, differences occur between the individual in-situ



reference sensors including distinct different slopes of the fitted hyperbolic regression model as well as larger deviations from the model fit indicating different soil moisture dynamics at the individual reference sensor locations.

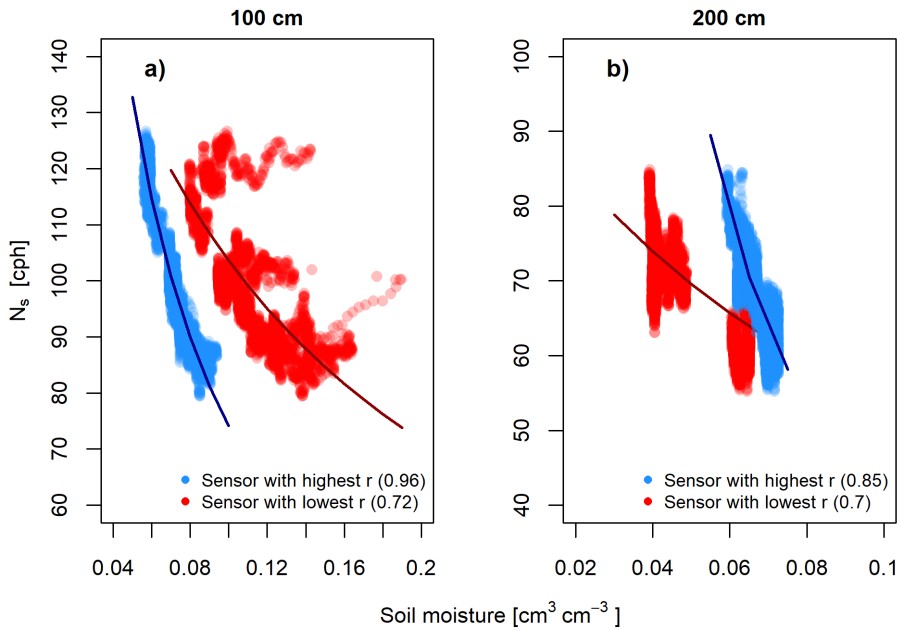

**Figure 8.** Comparison of the hourly corrected neutron intensity $N_s$ smoothed with a 25 h moving average and values from the reference soil moisture sensors with the highest and lowest Pearson's correlation coefficient r between the observed data and predicted data from a hyperbolic non-linear least squares fit model in the form of Eq. (5). (a) at 100 cm depth and reference sensors in 70 and 130 cm depth (b) at 200 cm depth with the corresponding reference soil moisture sensors also at 200 cm depth.

### 3.2.2 Predicting neutron ratios from reference soil moisture observations

In order to provide experimental proof of the proposed downhole application of cosmic-ray thermal neutron sensing (d-CRNS), in a first step, observed soil moisture time series along the different reference sensor profiles are used to predict $N_r$ based on Eqs. 5–9. Again, for the measurement depth of 100 cm, we include the sensor profiles with sensors down to 70 and 130 cm depth as there are no reference sensors available at the same measurement depth. For each sensor profile, the average soil moisture content is calculated from all sensors along the profile weighted by the depth range covered by each sensor down to

the maximum depth the respective sensor profile (70 and 130 cm) to calculate $\hat{\theta}_{SM}$. Additionally, the soil moisture time series from the sensors installed at depths of 70 and 130 cm are defined as $\theta_{SM}$ for each individual profile depending on the maximum profile depth. This leads to the set of predicted $N_r$ time series shown in Fig. 9. The observed time series of $N_r$ of the CRNS lies within the range of predicted $N_r$ time series although the values observed at 100 cm depth are slightly shifted towards the range of the the neutron ratios predicted from reference sensor profiles with a depth of 130 cm. Furthermore, the dynamics of the



predicted neutron ratios from reference soil moisture sensors match the dynamics of the CRNS-based values, which becomes especially visible during the rainfall event in the beginning of November 2021 (Fig. A1) as well as during the period in March 2022 where only very little rainfall was observed. The latter led to a decrease of soil moisture and hence, an increase of the observed $N_r$. Additionally, the short-term neutron ratio variations of the observed time series $N_r$ are strongly reduced when the time series is smoothed with a 49 h compared to a 25 h moving average which better corresponds to the $N_r$ time series

calculated from reference soil moisture sensors.

Similarly, a set of $N_r$ time series is calculated from the available reference soil moisture sensor profiles with sensors in depths down to 200 cm and 450 cm. At a measurement depth of 200 cm reference measurements are available in the exact depth of the detector tube location while the neutron ratios observed at 500 cm are compared to those predicted from sensor profiles with a maximum sensor depth of 450 cm. Fig. 9c shows the observed and predicted time series of $N_r$ in 200 cm. The temporal

dynamics of the predicted $N_r$ time series are smaller compared to those predicted from soil moisture sensors in shallower depths. This matches the dynamics of the observed $N_r$, however, stronger short-term fluctuations become more visible here. Although the dynamics are dampened, in both, the observed and the different predicted $N_r$ time series, the soil moisture increase caused by the intense rainfall event in late August 2021 is clearly visible (Fig. A2). In contrast, the predicted $N_r$ time series from soil moisture sensors down to 450 cm depth do not show any dynamics over the measurement period (Fig. 9d). However, despite

the short-term fluctuations visible in the observed neutron ratio, no trend can be observed which is in line with the predicted values of $N_r$. In both, the measurement depth of 200 cm and 500 cm, the observed time series of $N_r$ largely lies within the set of time series predicted with different reference soil moisture sensor profiles.

### 3.2.3 Estimating soil moisture from neutron ratio observations

The soil moisture time series of $\theta_{SM}$ derived from the observed $N_r$ in 100 and 200 cm are shown in Fig. 10. The estimated soil

moisture time series follow the general dynamics of the reference values of $\theta_{SM}$ in both depths. During the observed intense precipitation events in August and November 2021, both $\theta_{SM}$ in 100 and 200 cm show a distinct increase, which can also be seen in the soil moisture time series observed by the reference sensors. Similarly, the dry period in March 2022 results in a decrease of soil moisture indicated by $\theta_{SM}$ estimated from $N_r$ in 100 and 200 cm as well as by the in-situ reference sensors in the respective depths. While the absolute values of $\theta_{SM}$ in 100 cm lie in the range of observed soil moisture values from

the different in-situ reference sensors available, the values of $\theta_{SM}$ in 200 cm are at the upper end of the set of time series of reference sensors.

## 4 Discussion

### 4.1 Feasibility assessment

The particle transport simulations conducted in the scope of this study revealed a distinct relationship of the neutron ratio

($N_r$) with the local soil moisture content as well as with the shielding depth. As a consequence, changes in both the local soil

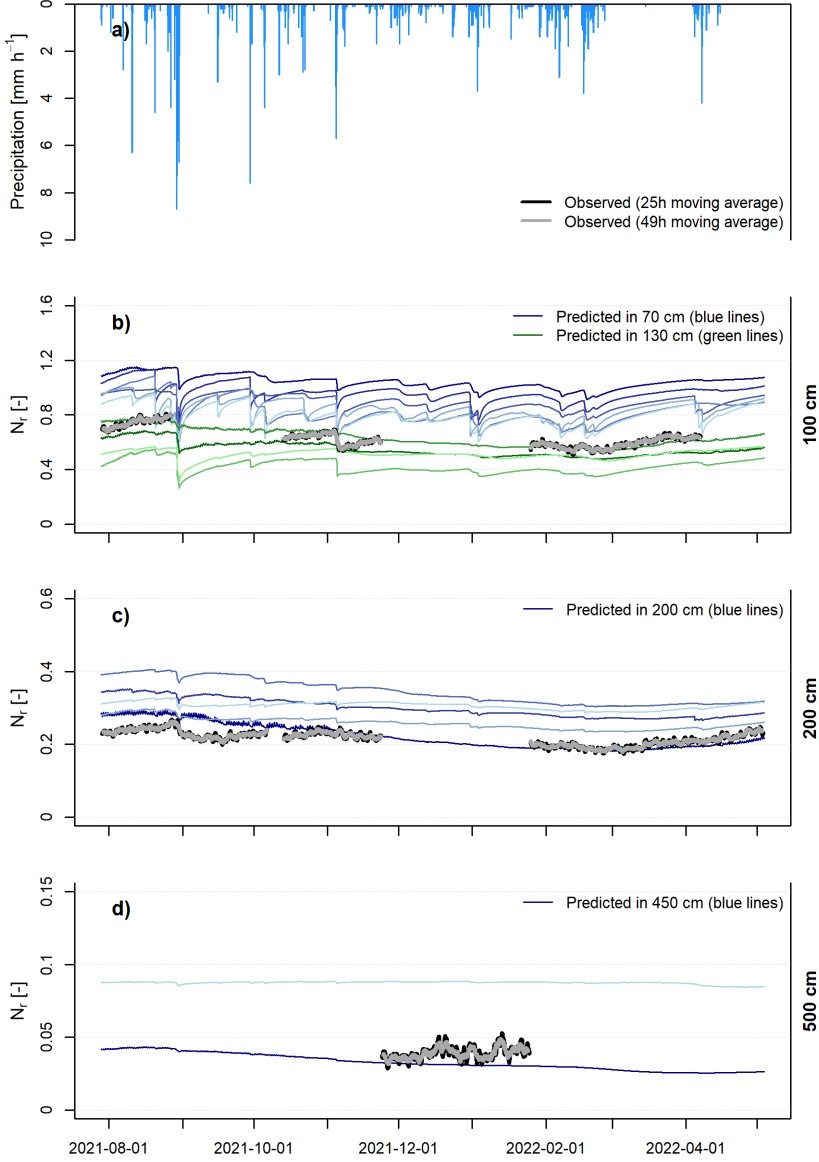

**Figure 9.** (a) Hourly precipitation observed at the study site during the experiment. (b)-(d) Observed (by CRNS detector at 100, 200 and 500 cm depth) and predicted (based on reference soil moisture measurements in similar depths) time series of $N_r$. The different times series of the predicted $N_r$ in each depth represent the results for every individual soil moisture sensor in that depth ($\theta_{SM}$) and the average from the associated sensor profile from the soil surface to the depth of the CRNS ($\hat{\theta}_{SM}$).

moisture content in the depth of measurement and in the average soil moisture content above the detector (due to its contribution

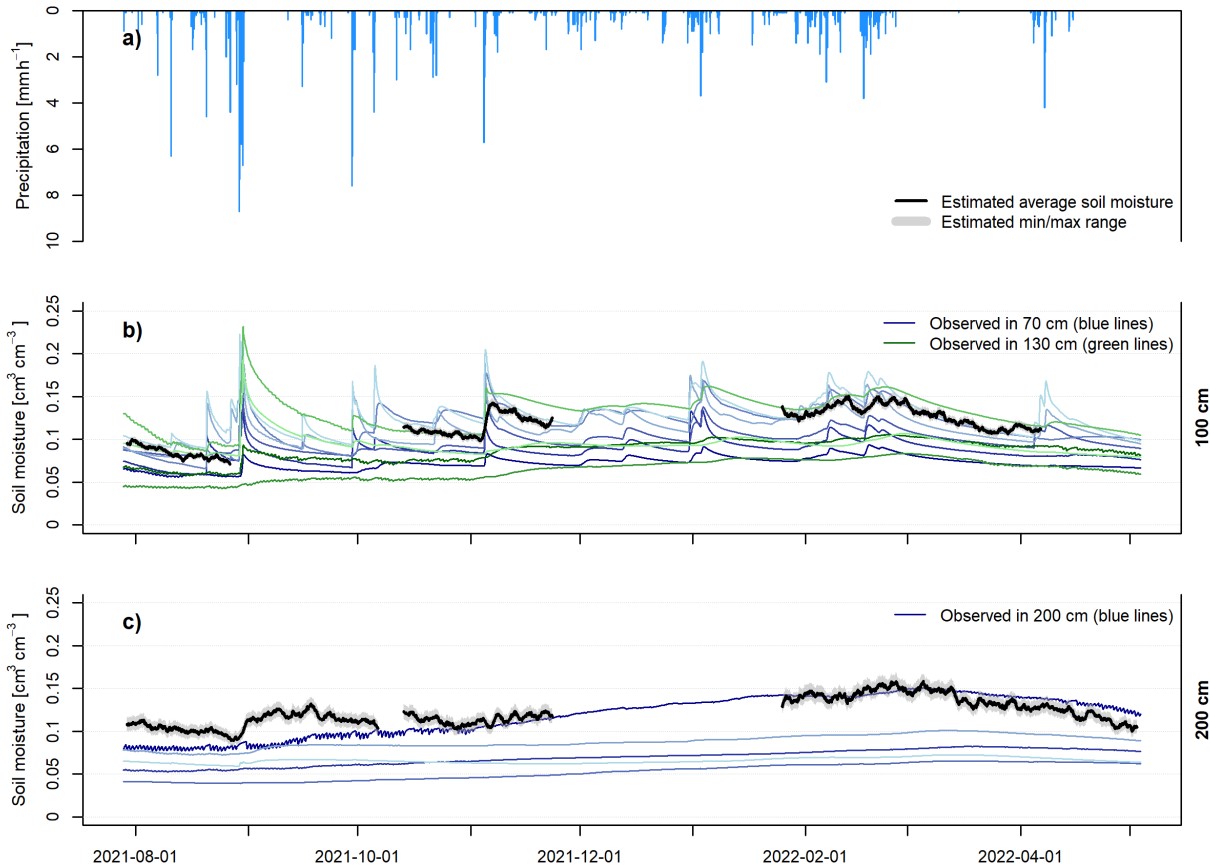

**Figure 10.** The observed time series of $\theta_{SM}$ from reference soil moisture sensor profiles and estimated time series of $\theta_{SM}$ from $N_r$. The different colours indicate the observed time series from individual reference sensors in the respective depth. In both depths, corrected neutron intensities were smoothed with a 49 h moving average prior to calculating $N_r$ and subsequently estimating soil moisture. (a) Hourly precipitation observed at the study site during the experiment. (b) $\theta_{SM}$ from reference soil moisture sensors in 70 and 130 cm depth and estimated time series of $\theta_{SM}$ in 100 cm. (c) $\theta_{SM}$ from reference soil moisture sensors and estimated time series of $\theta_{SM}$ from $N_r$ in 200 cm depth.

to the shielding depth) alter the neutron intensity and, thus, $N_r$ observed by a downhole neutron detector. This illustrates the general possibility of deriving soil moisture information from $N_r$ observed in the scope of d-CRNS.

Before neutron ratios can be calculated from downhole neutron intensities and from neutron intensities observed with the
same detector above a water surface, the latter need to be corrected for variations in absolute humidity. The response of thermal neutrons to changes in absolute air humidity found here is less than half of the value which was reported by (Rosolem et al., 2013) and may be explained by the generally smaller response of thermal neutrons to changes in hydrogen (e.g., Weimar et al., 2020). The rate of change between the thermal neutron intensity and absolute humidity derived in our study can be used to adjust the correction approach described by (Rosolem et al., 2013) for epithermal neutrons so that it can be used to correct





thermal neutrons observed above water instead. Although this correction approach may also be used as a first approach to correct thermal neutron intensities measured above soils, it should be noted that the response of neutron intensity with absolute humidity may change with soil moisture content, as reported for epithermal neutrons by (Köhli et al., 2021). This illustrates the need for developing more sophisticated approaches for correction thermal neutron intensities for variations in absolute air humidity.

A specific feature observed in the neutron transport simulations is that while neutron intensities generally increase with decreasing shielding depths, we find a maximum of the neutron intensity and hence, neutron ratio for low soil moisture content in shielding depths between 75 and $100\,\mathrm{g\,cm^{-2}}$, with lower neutron intensities at shielding depths below $75\,\mathrm{g\,cm^{-2}}$. A maximum secondary neutron intensity in shallow depths below the surface has been simulated for e.g. the Earth's (Phillips et al., 2001; Zweck et al., 2013) and the Mars' surface (Zhang et al., 2022) and is linked to production of neutrons in the

upper soil layers through nuclear evaporation as well as moderation by elastic scattering and absorption processes. Based on the neutron transport simulations conducted for a soil bulk density of $1.43\,\mathrm{g\,cm^{-3}}$, an intensity maximum occurs in the soil up to a soil moisture content of $0.045\,\mathrm{cm^3\,cm^{-3}}$ at a shielding depth of $100\,\mathrm{g\,cm^{-2}}$ (Fig. 3). At higher soil moisture contents the maximum disappears. This may be linked to a smaller leakage of neutrons to the atmosphere as more hydrogen causes more elastic scattering in the soil.

The tube and filling materials of the groundwater well noticeably influence the signal of the active neutron probe (e.g., Keller et al., 1990) due to e.g. an influence of material chemistry on thermal neutron intensities (e.g., Quinta-Ferreira et al., 2016). The simulation results of the present study show that the dimensions of the sphere of influence can be comparable for a well tube made of PVC and stainless with equal material thickness. As the average difference for $R_{86}$ between a stainless steel and a PVC tube is approximately 1 cm and for $Vc_{86}$ as well as $V_{86}$ between 8 and 5 cm, the influence of the material composition

on the sphere of influence can be identified but may be regarded as negligible. However, a thinner or thicker wall of the tube may lead to different results. In contrast, the neutron ratios differ between a well tube composed of stainless steel and PVC. Although the response to changes in soil moisture are similar, the absolute neutron ratios for a stainless steel tube of equal thickness are noticeably lower. This may be attributed to the influence of the higher absorption probability of chlorine for thermal neutrons in the PVC material. For the same reason, a thinner PVC material reduces the absolute neutron ratios to a

smaller degree compared to a thicker PVC wall tubing.

    For similar reasons, the effect of filling material surrounding the actual tube of the groundwater observation well might be of importance in the scope of d-CRNS and should be assessed in future research. However, at our study site the width of the filling material around the tube was only 10 cm and the filling material was similar to the original material of surrounding, undisturbed soils: A sand filling for soil layers composed of sandy soils and clay for less conductive layers in greater depths.

Similar soil moisture dynamics in the refilled material compared to the undisturbed material can therefore be assumed for our test site.

    The radius ($R_{95}$) of the sphere of influence of the d-CRNS approach, even at high soil bulk densities, is larger compared to active neutron probes. For the latter, neutrons have to traverse the soil volume twice, on the way into the soil and back to the detector. In contrast, for passive applications, the water-sensitive neutrons in the fast energy range are directly generated in





the soil by high-energy neutrons, protons and muons and only have to traverse the soil volume once. Consequently, secondary cosmic-ray neutrons can reach the downhole neutron detector from origins in greater distances.

According to the particle transport simulation, the most sensitive volume lies above the detector center. This is related to the source of cosmic-ray neutrons from above the soil surface. Similarly to the measurement footprints of above-ground CRNS, the sphere of influence varies with soil water content and with bulk density, with a higher sensitivity close to the neutron

detector. As a consequence, reference soil moisture measurements close to the detector are likely to be more important than those in greater distances of the integration volume when predicting neutron ratios from reference soil moisture observations. Thus, further research may be required in order to assess whether weighting schemes for reference soil moisture measurements similar to those developed for above-ground CRNS are necessary in order to improve predicted neutron ratios. It should be noted that the simulated footprint dimensions are only valid for the modelled detector geometry and may vary with detector

size and well or access tube dimensions.

Thermal neutrons detected with an unshielded, bare neutron detector as it is used in the present study are more sensitive to absorption processes compared to neutrons in the epithermal energy range which are dominated by moderation processes. As a consequence, soil chemistry influences the observed neutron intensity in the soil and hence, also the derived neutron ratios from thermal neutrons and the neutron ratio variations with changes in soil moisture contents due to differing nuclear

absorption probabilities in soils with different chemical compositions (e.g., Zreda et al., 2008; Quinta-Ferreira et al., 2016). However, we purposely used a simple soil chemistry setup in the particle transport simulations which has been used as a the standard configuration in several simulation studies (e.g., Köhli et al., 2015, 2021). This is done in order to derive a first set of equations describing the neutron ratio response and sphere of influence in the scope of the d-CRNS approach which can be applied over wide range of observation sites instead of tailoring the simulation setup and thus, the derived equations

specifically to the observation site of this study. Although a standard soil chemistry was used to derive the transfer functions, the observed neutron ratios match the dynamics and ranges of predicted neutron ratios from in-situ reference sensors indicating the suitability and applicability of the d-CRNS approach as well as the equations derived at this site. A different soil chemistry may only introduce an overall damping of the measured intensity (Köhli and Schmoldt, 2022). Nonetheless, the conclusions are limited by the single site chosen for this study. Further research is required to test and validate the transferability of the

approach and to investigate the influences of, e.g., varying soil chemical compositions, access tube and filling materials as well as suitable technical set-ups for practicable applications.

## 4.2   Uncertainties

The experimental setup of field measurements conducted in the scope of this study comprised the measurement of thermal neutrons with an unshielded proportional detector in 100, 200, 500 and 1000 cm depth with colocated reference in-situ soil

moisture sensors installed down to 450 cm depth. Observed neutron intensities in the groundwater observation well show distinct response with changing soil moisture contents in the depth of measurement, indicating the possibility of measuring soil moisture and supporting the results from the various particle transport simulation scenarios.





In line with the exponential decrease of the absolute neutron flux with increasing soil depth, the uncertainty in the neutron intensity as well as the neutron ratio ($N_r$) increases. In general, the observed downhole neutron intensities are lower than observed above a water surface and thus, lower compared to intensities expected for above-ground CRNS applications. As the uncertainty increases with decreasing neutron intensity, the hourly time series needs to be averaged over longer time intervals compared to time series of above-ground neutron detectors. While above-ground neutron time series are typically averaged with a moving average of 13 to 25 h (e.g., Bogena et al., 2013; Schrön et al., 2018), for d-CRNS longer moving average windows of 25 to 49 h are more suitable. However, with respect to the passive, continuous nature of d-CRNS as well as the expected smaller soil moisture dynamics in greater depths where also soil moisture responses are more strongly dampened, larger averaging intervals are acceptable.

Additional improvements can be made to reduce the uncertainty in observed downhole neutron time series. For example, detector tube no. 2 showed significantly lower neutron intensities compared to tube no. 1 which can be related to settings of the instrument electronics. As the detector system used in this study was reassembled from different CRS1000 neutron detector systems, the neutron pulse module settings of tube no. 2 did not match the ideal configuration for the proportional counter tube attached and thus, a large part of potentially countable thermal neutrons were discarded. This led to the lower observed intensities.

In spite of these uncertainties, our study reveals that observed $N_r$ follow the temporal dynamics of predicted $N_r$ from Eqs. (5-9) and lie within the range of predicted $N_r$ values from in-situ reference soil moisture sensor profiles. Both, intense rainfall events and gradual soil moisture changes during drying periods could be observed in the downhole measurements. The measurements in 100 cm depth exhibit stronger dynamics compared to those in 200 cm which is in line with reference soil moisture time series and predicted neutron ratios. The observed $N_r$ time series at 100 cm is closer to the one predicted for 130 than for 70 cm depth which may be explained by the fact that the predicted time series of $N_r$ strongly differ between 70 and 130 cm depth but also among the sensors within the two depth layers. This is due to markedly different values and dynamics of the individual soil moisture sensors. This marked variability of point-scale soil moisture hampers the direct comparison to the results derived from d-CRNS. It should be noted that especially in large depths of 500 cm only very little soil moisture dynamics occur while the d-CRNS uncertainty is high, limiting the range of suitable application depths of d-CRNS. The results of this initial study revealed that the predicted hourly time series of $N_r$ from reference soil moisture sensors in 70 and 130 cm have a coefficient of variation of 5 to 12 % for the study period. In 200 cm, the coefficients of variation are in a range between 3 and 16 %. The time series of the observed $N_r$ need to be smoothed with a 49 h moving average to suppress noise and to result in coefficients of variation in the same order of magnitude, i.e., a value of 11 % in 100 cm and 9 % in 200 cm depth, respectively. In contrast, the coefficient of variation of the observed $N_r$ in 500 cm depth with the same moving average applied is 4.3 times larger than the maximum coefficient of variation from the predicted $N_r$ time series in 450 cm depth. According to these findings, the d-CRNS observations can be expected to be dominated by noise at the depth of 500 cm, unable to resolve the small soil moisture variations at this depth. However, the d-CRNS approach may be suitable for resolving the soil moisture dynamics at this site for shielding depths up to at least 330 g cm$^{-2}$ which roughly corresponds to a soil depth of at least 200 cm, when a moving average interval of 49 h is applied.





The uncertainties of $\theta_{SM}$ that are caused by the simplified estimation method used here (section 3.1.3) are comparatively small. Although we allowed the assumed mean soil moisture in the unsaturated zone above the sensor to vary between wilting
point and field capacity for estimating the soil moisture in the sensor depth (which represents the upper bound of the possible uncertainty), the resulting uncertainty bounds of $\theta_{SM}$ are very small and hardly relevant for the depth of 100 cm, and still small for the depth of 200 cm (see Figure 10).

While it is a major advantage of this study that in-situ point-scale soil moisture observations for evaluating the d-CRNS approach are available at the study site at 200 cm and even 450 cm depth, all reference sensors are unfortunately located
outside the sphere of influence of the downhole neutron detectors in the groundwater observation well (at distances between 20 and 40 m). However, the observed $N_r$ lies within the set of predicted time series of $N_r$ from in-situ reference sensor profiles and follows the general temporal dynamics the predicted time series of $N_r$ which supports the applicability of d-CRNS.

A key motivation of this study is to provide a new methodological approach to derive soil moisture information from deeper layers of the vadose zone in a larger integration volume compared to point-scale in-situ sensors. However, deriving soil moisture
from observed $N_r$ is difficult, as two soil moisture variables influence the latter: the soil moisture content in the measurement depth $\theta_{SM}$ and the average soil moisture from soil surface to the detector center $\hat{\theta}_{SM}$. A first option would be the use of Eq. (5-9) as a forward operator in combination with a soil hydraulic model. Similar approaches have been conducted using e.g. the COSMIC forward operator model code (Shuttleworth et al., 2013) for above-ground CRNS applications (e.g., Brunetti et al., 2019; Barbosa et al., 2021). Although the application as a forward operator in combination with soil hydraulic modelling may
produce more accurate results as the soil water transport is simulated in different depths and also allows for the retrieval of soil moisture simulated in several soil layers, a large number of input parameters is required which may not be available at all sites. In contrast, the simple approach to estimate the local soil moisture content in the depth of measurement as the most sensitive variable showed that the resulting soil moisture time series follow the dynamics and also lie in the range of expected values derived from in-situ soil moisture sensors. However, it should be noted that this approach may be less accurate and only allows
for an estimation of the local soil moisture time series.

As this study is restricted to a single observation site, further research is required to test both the soil hydraulic model-based approach and the approach used here under different site-specific boundary conditions, set-ups and measurement depths. Nevertheless, these two approaches are available in order to retrieve soil moisture information from d-CRNS and hence, allowing for direct application of the methodological approach in the scope of hydrological research.

**5  Conclusions**

In this study we tested the feasibility of CRNS downhole applications to estimate soil moisture at greater depth by combining particle transport simulations with a first application in the field. Although using an unshielded neutron detector most sensitive to thermal neutrons, a distinct response to changes of the soil moisture content in the observation depth as well as in the shielding depth above the neutron detector was found. This illustrates the possibility of observing soil moisture values in greater
depth with d-CRNS without additional soil moisture information for calibration. This is achieved through the calculation of





neutron ratios using a measurement above water. The sphere of influence has a unique shape differing from those expected for active neutron probes as the neutron source and detector are not collocated. As detected neutrons are produced directly in the soil, the sphere of influence is much larger compared to an active neutron probe and thus, d-CRNS allows for deriving representative average soil moisture information in different depths of the root-zone.

Our measurements of downhole neutron intensities and calculated neutron ratios from a groundwater observation well provide experimental evidence that downhole thermal neutron detectors are sensitive to changes in soil moisture contents in the measurement depth. Simultaneously, the results of this study illustrate the opportunity of using existing monitoring infrastructure to retrieve soil moisture information from deeper soil layers. The transfer functions developed from particle transport simulations in the scope of this study can be used as an forward operator to calculate neutron signals from soil moisture infor-

mation. In combination with soil hydraulic models, the forward operator can be then used to derive soil moisture contents in future applications. When the use of complex models is hampered by e.g. scarce data, a simple approach can be used for a first estimation of the soil moisture in the measurement depth.

In conclusion, we provide both, simulation-based and experimental evidence for the feasibility of using downhole secondary cosmic-ray neutrons for the continuous, non-invasive estimation of soil moisture from greater depth. This method has several

advantages compared to traditional in-situ soil moisture sensors: the larger integration volume of the measurement counteracts the usual problems caused by the high spatial variability of soil moisture even at small scales as a result of the subsurface heterogeneity. It furthermore does not require demanding installation procedures by simply using existing infrastructure (i.e. observation wells) which is readily available in many locations as part of standard monitoring networks. The mathematical relationships presented allow for predicting the neutron signal from soil moisture information and approaches are available for

deriving soil moisture contents from downhole neutron observations. However, as this study poses several limitations and is only a first proof-of-concept, further testing and developments will be necessary. This effort is worthwhile, especially since deep soil moisture measurements become more and more important to monitor subsurface droughts or water stress in forests, and to validate hydrological models and extrapolation efforts from remote sensing products.





*Data availability.* All data sets are available from the authors upon request.

**Appendix A**

**Table A1.** Sensor distribution of the reference soil moisture sensor profiles at the study site located at a distance of about 20 to 30 m from the groundwater observation well.

| | | | | Profile no. | | | |
|---|---|---|---|---|---|---|---|
| Depth [cm] | 1 | 2 | 3 | 4 | 5 | 6 | 7 |
| 10 | ✓ | ✓ | ✓ | ✓ | ✓ | ✓ | ✓ |
| 20 | ✓ | ✓ | | ✓ | ✓ | ✓ | ✓ |
| 30 | ✓ | ✓ | ✓ | ✓ | ✓ | ✓ | ✓ |
| 50 | ✓ | ✓ | ✓ | ✓ | ✓ | ✓ | ✓ |
| 70 | ✓ | ✓ | | ✓ | ✓ | ✓ | ✓ |
| 130 | | ✓ | ✓ | | | ✓ | ✓ |
| 200 | ✓ | | | ✓ | ✓ | ✓ | ✓ |
| 300 | | | | | | ✓ | ✓ |
| 450 | | | | | | ✓ | ✓ |

**Table A2.** Fitted parameters for Eq. (2-7) derived from particle transport simulation scenarios for a stainless steel well tube with a wall thickness of 7.5 mm.

| Eq. no. | Variable | $p_1$ | $p_2$ | $p_3$ | $p_4$ | $p_5$ | $p_6$ | $p_7$ | $p_8$ | $p_9$ | $p_{10}$ |
|---|---|---|---|---|---|---|---|---|---|---|---|
| (2) | $R_{86}$ | 185 cm | 0.164 | 4.51 cm | 2 | | | | | | |
| (3) | $V_{86}$ | 115 cm | 1.2 | 0.211 | 23.9 cm | 0.5 | | | | | |
| (4) | $Vc_{86}$ | 31.2 cm | 4 | 6.7 cm | 0.6 | | | | | | |
| (6) | $F_1$ | 0.502 | -0.0206 cm² g⁻¹ | 0.0158 | -0.000839 cm² g⁻¹ | 0.531 | -0.00674 cm² g⁻¹ | | | | |
| (7) | $F_2$ | | | | | | | 0.0667 | 0.000139 cm² g⁻¹ | 0.435 | -0.0172 cm² g⁻¹ |

**Table A3.** Fitted parameters for Eq. (6-7) derived from particle transport simulation scenarios for a PVC well tube with a wall thickness of 5 mm.

| Eq. no. | Variable | $p_1$ | $p_2$ | $p_3$ | $p_4$ | $p_5$ | $p_6$ | $p_7$ | $p_8$ | $p_9$ | $p_{10}$ |
|---|---|---|---|---|---|---|---|---|---|---|---|
| (6) | $F_1$ | 0.348 | -0.0206 cm² g⁻¹ | 0.011 | -0.000839 cm² g⁻¹ | 0.369 | -0.00674 cm² g⁻¹ | | | | |
| (7) | $F_2$ | | | | | | | 0.0482 | 0.000139 cm² g⁻¹ | 0.314 | -0.0172 cm² g⁻¹ |

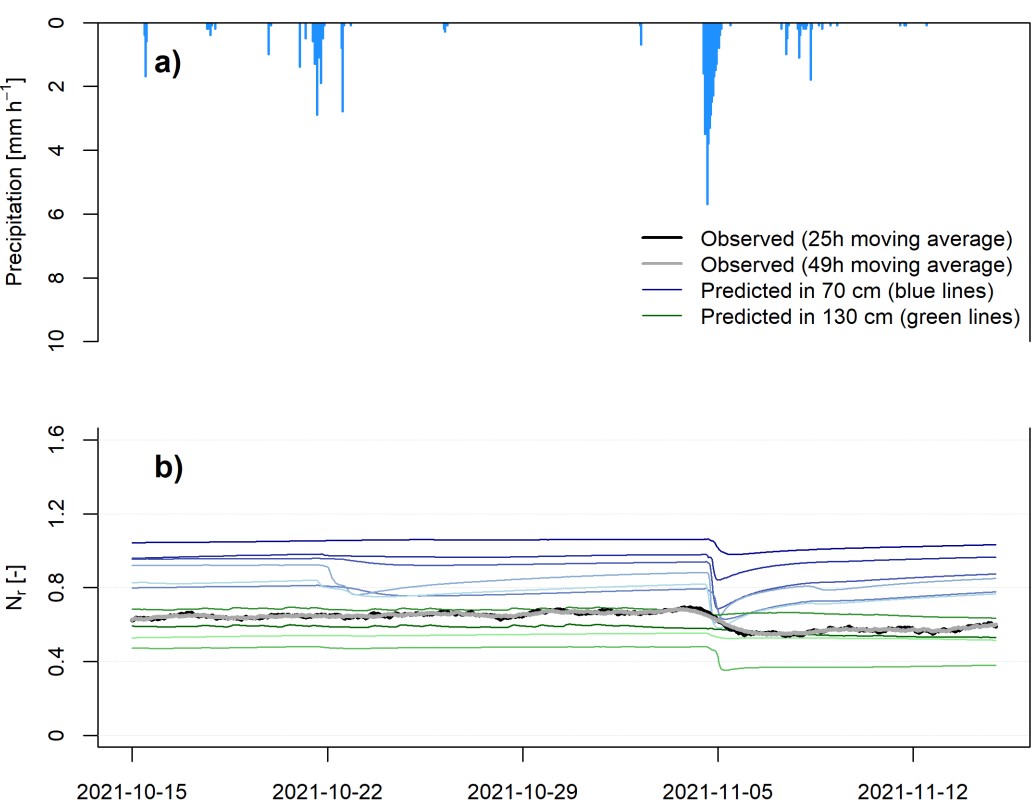

**Figure A1.** (a) Hourly precipitation observed at the study site during a detail period of the experiment in October/November 2021 as well as (b) the observed time series of $N_r$ in $100\,\text{cm}$ and predicted time series of $N_r$ from reference soil moisture sensor profiles in 70 and $130\,\text{cm}$ depth. The different colours indicate the predictions from individual reference sensor profiles.





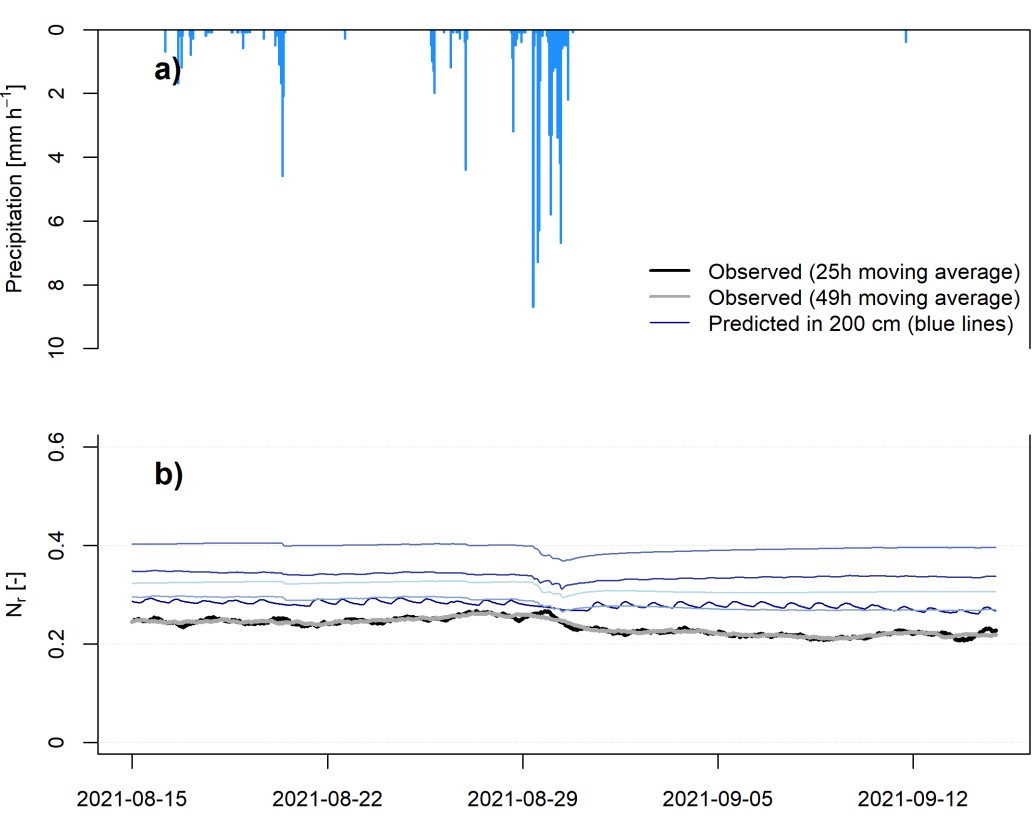

**Figure A2.** (a) Hourly precipitation observed at the study site during a detail period of the experiment in October/November 2021 as well as (b) the observed time series of $N_r$ in 200 cm and predicted time series of $N_r$ from reference soil moisture sensor profiles in 200 cm depth. The different colours indicate the predictions from individual reference sensor profiles.



*Author contributions.* DR had the idea for this study, designed the experiment, performed the data analysis and wrote the manuscript. JW conducted and analysed the particle transport simulations and wrote the manuscript. MK and MS assisted in performing and analysing neutron transport simulations and provided valuable ideas regarding the manuscript. TB and AG designed the soil moisture and groundwater monitoring network and contributed to the writing of the manuscript. MM assisted in planning and designing the experimental set-up as well
as conducting the field measurements.

*Competing interests.* Markus Köhli and Jannis Weimar hold a CEO position at StyX Neutronica GmbH. Theresa Blume is Chief Executive Editor of the journal HESS.

*Acknowledgements.* This study was conducted as part of the research unit CosmicSense funded by the German Research Foundation (Deutsche Forschungsgemeinschaft, DFG-FOR2694, project no. 357874777). We gratefully acknowledge the technical support of Jörg
Wummel and Stephan Schröder who maintain the observation sites in TERENO-NE funded by the Helmholtz Association. In addition, we would like to thank Paul Voit for his assistance in data acquisition, field and laboratory work. Further, we would like to thank the Müritz National Park for the continuing support and collaboration. Lastly, we acknowledge the NMDB database (www.nmdb.eu) founded under the European Union's FP7 programme (contract no. 213007), and the PIs of individual neutron monitors for providing data.





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
