# Peer review of "A change in perspective: Downhole cosmic-ray neutron sensing for the estimation of soil moisture"

_Hydrology and Earth System Sciences, 2022_

## Author Response (AR1)

**Revision of the manuscript "hess-2022-364"**

This document contains a point-by-point reply to the comments of all reviewers, corresponding author responses and adjustments made in the original manuscript.

The major comments of the three anonymous reviewers concerned the length of the manuscript on the one hand and, on the other hand, required additional information regarding the derivation of soil moisture from observed downhole neutron ratios, information on the role of different high-energy particles in generating water-sensitive neutrons in the soil as well as addressing further sources of uncertainty in the described approach. We added and modified paragraphs throughout the manuscript in order to clarify aspects to the reader and improve the overall manuscript.

The reviewer comments are denoted with R1-1 (reviewer 1, comment 1) and A1-1 (author response to reviewer 1, comment 1). When changes are made to phrases in the original manuscript, we state the original and adjusted sentence/paragraph or provide the phrase added.

**Reviewer #1:**

We thank reviewer #1 for taking the time to review our manuscript. In the following section we will reply to all comments of reviewer #1 with R1-1 (i.e. reviewer 1, comment 1) and A1-1 (i.e. author response to R1-1), respectively.

**R1-1:** It is intresting about the d-CNRS for deep soil water measurement through groundwater wells, but I only suggest the manuscript should be shorted and the story wil be easy for reading..

**A1-1:** We are glad that the downhole CRNS approach appeals to the reviewer. We know that the manuscript is detailed and
therefore a bit long. Given that this is the first study that presents d-CRNS, we decided to provide both, a broad as well as an in-depth analysis and explanation of the d-CRNS approach. This includes necessary neutron transport simulations and field measurements to understand, apply, and reproduce the method.

In our opinion, this requires:
-   a theoretical description of the measurement volume (i.e., the sphere of influence) and the response of downhole neutron intensities in different shielding depths based on neutron transport simulations
        -   the derivation of equations for a mathematical description of the sphere of influence and a first set of equations to describe the response of neutron intensities to changes in soil moisture in different (shielding) depths
        -   a recommendation for an inversion algorithm to derive soil moisture from observed downhole neutron intensities
-   a description of correction procedures for downhole neutron observations which are not identical with those for above-ground neutron intensities and lake-side neutron measurements to facilitate calibration against in-situ reference measurements of soil moisture
        -   and lastly, a first field application of the d-CRNS approach to demonstrate its applicability

Addressing these points leads to the slightly extended length of the manuscript. As stated by the other two reviewers, additional information should be added to the manuscript which makes the shortening of the manuscript difficult. In order to reduce the length of the main text, we decided to shift the section (lines 362-370) on describing the applied methodology to derive soil moisture from observed downhole neutron ratios to the appendix of the manuscript.

**Reviewer #2:**

We thank reviewer #2 for taking the time to review our manuscript. In the following section we will reply to all comments of reviewer #2 with R2-1 (i.e. reviewer 2, comment 1) and A2-1 (i.e. author response to R2-1), respectively.

**R2-1:** It would be good to see the estimated soil moistures from above ground CRNS in Figure 10. How different the values measured by above ground CRNS and downhole CRNS at 100 m depth?

**A2-1:** We agree that a comparison with the soil moisture time series derived from above-ground CRNS would be interesting. For the sake of readability of Figure 10, we added the following additional figure to the appendix of the manuscript showing
the soil moisture time series derived from above-ground CRNS as well as from downhole CRNS in 100 cm depth and 200 cm depth:

[Figure]

Fig. X: Hourly precipitation time series and (b) the different soil moisture time series derived from CRNS. The neutron
observations from above-ground CRNS was processed with standard correction and calibration procedures (site Serrahn, Bogena et al. 2022). A 25h-moving average was applied to the corrected neutron intensities prior to deriving soil moisture from above-ground CRNS observations with the standard transfer function (Desilets et al. 2010, Köhli et al. 2021). Marked periods with snow cover represent periods with fractional to full snow cover and snow depths up to 10-15 cm.

**R2-2:** Is the precipitation in liquid form during the winter? Does snow affect the downhole neutron intensities?

**A2-2:** This is an important point and we are glad that the reviewer mentioned this aspect as this is not included in the manuscript yet. We will correct this in the revised manuscript. At our study site, most precipitation in winter occurs as rain and only rarely do we have a thin snow cover for a few days.

In general, liquid precipitation is assumed to infiltrate directly and then influences downhole neutron intensities as an increase in soil moisture by its contribution to shielding depth, i.e. the total amount of mass above the downhole neutron detector. This is described by equation 9.

Like any other mass above the neutron detector, a snow cover also contributes to shielding depth and is not considered in equation 9 yet. Consequently, we changed the term describing above-ground wet biomass $D_{AGB}$ to $D_{AGM}$ now describing all additional above-ground mass on top the soil including above-ground wet biomass and the water equivalent of a snow cover in g cm$^{-2}$. The description in the text has been adjusted accordingly.

As stated in the manuscript, average soil moisture changes from the surface to the depth of measurement through their contribution to shielding depth are much less influential on downhole neutron intensities than soil moisture changes in the depth of measurement. Thus, small amounts of snow are likely to have a rather negligible effect on downhole neutron intensities. In contrast, at study sites with thick snow covers a distinct effect can be expected.

We thank reviewer #3 for taking the time to review our manuscript and the helpful comments which will improve the quality of the manuscript. In the following section we will reply to all comments of reviewer #3 by numbering reviewer comments starting with R3-1 (i.e. reviewer 3, comment 1) and our response with A3-1 (i.e. author
response to R3-1).

**R3-1:** In the introduction (Chapter 1), the publication by Kodama et al. (1985, https://journals.lww.com/soilsci/Abstract/1985/10000/APPLICATION_OF_ATMOSPHERIC_NEUTRONS_TO _SOIL.1.aspx) should be mentioned.

This publication is about a passive below-ground neutron sensor and therefore it is needed to briefly mention how the current manuscripts relates to this work from the 1980's.

**A3-1:** We are glad about this suggestion and agree that it should be mentioned in the introduction of our manuscript.
Kodama et al. (1985) measure both, above-ground and below-ground cosmic-ray neutrons and describe the change of observed neutron intensity with varying soil moisture. The below-ground measurements of Kodama's study cover the shallow subsurface with depths down to 40 cm, using moderated detectors.

In these shallow depths, the maximum intensity of cosmic-ray neutrons can be expected, making it suitable for soil moisture estimation with comparatively high statistical accuracy. On the other hand, this shallow sub-surface
monitoring depth largely corresponds to the vertical measurement depth of above-ground cosmic-ray neutron sensing. In the scope of the present study, we explicitly explore the potential of CRNS beyond the measurement depths of classical above-ground CRNS. We also emphasize that placing detectors in shallow depths leads to the contribution of neutrons which previously interacted with the atmosphere and which originate from larger horizontal distances to the detector location. This makes it difficult to interpret the neutron signal and its response
to nearby soil moisture changes in the depth of the below-ground neutron detector. For this reason, the equations presented in this study are valid for shielding depths greater 75 g cm$^{-2}$. For example, a measurement depth of 40 cm with an average soil bulk density of 1.43 g cm$^{-3}$ would lead to a shielding depth of only 57 g cm$^{-2}$ for a completely dry soil.

In order to mention Kodama et al. (1985), we adjusted the following section starting in line 65:

Original:

"… as well as the typically non-continuous nature of snapshot measurement campaigns with active neutron probes. Against this background, we investigate the possibility of using CRNS-related sensors in a passive downhole technique (d-CRNS) to estimate soil moisture in different depths of the root zone and deeper unsaturated zone. "

Adjustment:

"… as well as the typically non-continuous nature of snapshot measurement campaigns with active neutron probes. Kodama et al. (1985) observed the response of cosmic-ray neutrons to changes in soil moisture in depths down to 40 cm., largely covering the sensitive measurement depth of above-ground CRNS.
Against this background, we investigate the possibility of using CRNS as a passive downhole technique (d-CRNS) to estimate soil moisture in different depths below 40 cm, also including the deeper unsaturated zone. For this, we installed CRNS neutron detectors in a standard groundwater observation well, and thus using the well casing above the groundwater level as an access tube."

**R3-2:** Chapter 2 Material and Methods, Section 2.4, page 9, lines 196-204: The HESS-reader could be helped by explaining a bit more how the roles protons, muons, and neutrons play, differ and how much different processes dominate in relative terms (moderation and in-soil neutron production)

Please, if explained in the manuscript text, keep this very brief.

**A3-2:** In order to give a brief example, we added the following sentence in line 200:

Original:

"Although simulating only neutrons and not e.g. protons and muons as well might be sufficient at the soil-atmosphere interface with a detector placed above the surface, the simulation of the neutron flux in different depths of the soil requires the inclusion of several other types of particles that induce neutron production in the soil volume itself. That is because the atmospheric neutron flux is attenuated strongly within the soil volume and in-soil neutron production dominates the thermal neutron flux below several decimetres soil depth."

Adjustment:

"Although simulating only neutrons and not also protons and muons might be sufficient at the soil-atmosphere interface with a detector placed above the surface, the simulation of the neutron flux in different depths of the soil requires the inclusion of several other types of particles that may induce neutron production in the deeper soil volume. As the atmospheric neutron flux is attenuated strongly within the soil volume the in-soil neutron production dominates the thermal neutron flux below several decimetres soil depth. In-soil neutrons are generated by different cosmic-ray particle species depending on the soil depth. Within the first few meters the inelastic collisions of high- energy protons and neutrons with atomic nuclei lead to particle production in hadronic showers (e.g. see Mollerach and Roulet 2018). During the collision neutrons are ejected from the nucleus with energies peaking at few hundred MeV (e.g. Gudima et al. 1983). The target nucleus remains in an excited state after the impact and deexcites via the emission of lower energetic neutrons with few MeV. This process is called evaporation. The hadrogenic neutron production falls off rapidly with soil depth due to the short penetration length of high-energy neutrons and protons.

Below that and down to several tens of meters, hadrogenic neutron production is significantly lower and is dominated by muons (Heusser 1996) through capture processes that releases neutrons with few MeV. "

**R3-3:** Chapter 3  Results, Section 3.1.3, page 7, line 362: Why from wilting point to field capacity (these are rather arbitrary soil moisture contents that, formally, relate to the plant grown) and why not to saturation?

**A3-3:** Section 3.1.3 describes an exemplary procedure to derive the soil moisture content in the depth of measurement from the presented equations and observed downhole neutron intensities. For the average soil moisture content in the layer between the surface and the depth of measurement, we argue that these boundaries are justified as it is unlikely that the average soil moisture content is larger than field capacity or reaches saturation over the entire depth range given the sandy soils and large groundwater table depth at the study site.

In contrast, in the depth of the d-CRNS measurements, soil water saturation may be reached and for this reason the upper limit is set to saturation in step 3 of the procedure (line 364).

However, the described approach to derive soil moisture in the depth of measurement remains an exemplary procedure and other boundaries might be used for different site conditions and may be tested in future studies. For this reason, we changed the following sentence in line 356:

Original:

"We propose an alternative, more simple approach to estimate the soil moisture time series in the depth of measurement from the observed neutron ratios by using Eq. (5-9)."

Adjustment:

"We propose an alternative, and more simple approach to estimate the soil moisture time series in the depth of measurement from the observed neutron ratios by using Eqs. (5-9). This approach is in some way exemplary in that while reasonable for the conditions of our study site, the assumed range of soil moisture between wilting point and field capacity in Eqs. 5-9 may need to be modified for other sites. "

**R3-4:** In chapter 4 Discussion: The possible uncertainties introduced by vertically highly heterogeneous soils should be discussed.

These heterogeneities can affect the outcomes through both vertically variant soil bulk densities and, during certain periods, vertically heterogeneous soil moisture contents.

To what extent does not knowing the exact vertical variation matter to the outcomes and uncertainty?

To what extent would vertically heterogeneous soils affect the applicability of the methods at different sites, given more in-situ sampled data might be needed?

**A3-4:** We agree with the reviewer that potential uncertainties arising from vertically heterogenous distributions of bulk density and soil moisture should be mentioned.

In fact, vertically distributed soil moisture information as well as bulk density profiles are essential parameters for the presented set of equations in order to predict downhole neutron ratios. The respective bulk density information is also required to estimate soil moisture contents in the depth of measurements following the procedure in chapter 3.1.3 making it necessary to consider uncertainties in these parameters. However, an in-depth assessment of the impact of the quality and uncertainties in vertically distributed soil moisture and bulk density information on derived neutron ratios and estimated soil moisture information is beyond the scope of this proof-of-concept study.

Considering the sensitive measurement volume of the downhole neutron detector, vertically heterogenous bulk density and soil moisture distributions may also be important. Similar to above-ground CRNS e.g. soil moisture close to the detector (centre) can be expected to have a larger impact on the observed neutron signal than the soil moisture in greater (vertical and horizontal) distance to the downhole neutron detector. This point as well as the need for further research has already been mentioned in lines 482-490.

Nevertheless, potential uncertainties of vertically highly heterogeneous soil bulk density and soil moisture distributions should be briefly mentioned in chapter 4.2 and thus, we adjusted the following paragraph (line 570):

Original:

"As this study is restricted to a single observation site, further research is required to test both the soil hydraulic model-based approach and the approach used here under different site-specific boundary conditions, set-ups and measurement depths.

Nevertheless, these two approaches are available in order to retrieve soil moisture information from d-CRNS and hence, allowing for direct application of the methodological approach in the scope of hydrological research."

Adjustment:

"As this study is restricted to a single observation site, further research is required to test both the soil hydraulic modelbased approach and the approach used here under different site-specific boundary conditions, set-ups and measurement depths. This also includes the consideration of uncertainties arising from soils with high vertical variability in bulk density (and possibly soil moisture), their impact on predicted neutron ratios, and on the estimated soil moisture in the depth of measurement. For example, a lower bulk density and a lower soil water content would lead to more neutrons penetrating into greater depths, and hence, to increased count rates and footprint volumes. Nevertheless, the two mentioned approaches are available for soil moisture retrieval from d-CRNS and could be applied under different soil-hydrological conditions in future studies."

**R3-5:** Chapter 4 Discussion, page 23, line 466 "...may lead to different results": such as?

**A3-5:** This statement refers to the comparison of the $R_{86}$, $Vc_{86}$ and $V_{86}$ of a tube made of PVC and stainless steel but with equal material thickness. A detector placed in a stainless steel or PVC tube with thinner walls is likely to have a larger measurement volume as less neutrons are absorbed or scattered away from the downhole neutron detector. Vice versa, a thicker wall can be expected to reduce the measurement volume. However, the degree of difference in the measurement volume depending on wall material and material thickness remains to be assessed and was therefore not specified further. We added the following statement (line 466):

Original:

"As the average difference for R86 between a stainless steel and a PVC tube is approximately 1 cm and for Vc86 as well as V86 between 8 and 5 cm, the influence of the material composition on the sphere of influence can be identified but may be regarded as negligible. However, a thinner or thicker wall of the tube may lead to different results.

In contrast, the neutron ratios differ between a well tube composed of stainless steel and PVC. Although the response to changes in soil moisture are similar, the absolute neutron ratios for a stainless steel tube of equal thickness are noticeably lower. This may be attributed to the influence of the higher absorption probability of chlorine for thermal neutrons in the PVC material. For the same reason, a thinner PVC material reduces the absolute neutron ratios to a smaller degree compared to a thicker PVC wall tubing"

Adjustment:

"As the average difference for R86 between a stainless steel and a PVC tube is approximately 1 cm and for Vc86 as well as V86 between 8 and 5 cm, a small effect of the material composition on the sphere of influence can be identified but may be regarded as negligible. However, a thinner or thicker wall of the tube is likely to have a stronger impact on the measurement volume. For instance, a thicker PVC wall can be expected to reduce the measurement volume.

In contrast, the neutron ratios differ between a well tube composed of stainless steel and PVC. Although the response to changes in soil moisture are similar, the absolute neutron ratios of a PVC well tube compared to a stainless steel tube of equal thickness are noticeably lower. This may be attributed to the influence of the higher absorption probability of chlorine for thermal neutrons in the PVC material. For the same reason, a thinner PVC material reduces the absolute neutron ratios to a smaller degree compared to a thicker PVC wall tubing."

**R3-6:** Chapter 5 Conclusions, page 27, lines 585-590: Please, discuss here and possibly in Chapter 4 Discussion, Section 4.2, that using soil hydraulic models could introduce further uncertainties due to assumptions made during the modelling process. In addition, applying a soil hydraulic model to obtain the results, make these less observations and more model results.

**A3-6:** We agree that such additional uncertainties should be briefly mentioned in the discussion chapter. We added the following sentence in line 566:

Original:

"A first option would be the use of Eq. (5-9) as a forward operator in combination with a soil hydraulic model. Similar approaches have been conducted using e.g. the COSMIC forward operator model code (Shuttleworth et al., 2013) for above-ground CRNS applications (e.g., Brunetti et al., 2019; Barbosa et al., 2021). Although the application as a forward operator in combination with soil hydraulic modelling may produce more accurate results as the soil water transport is simulated in different depths and also allows for the retrieval of soil moisture simulated in several soil layers, a large number of input parameters is required which may not be available at all sites."

Adjustment:

"A first option would be the use of Eq. (5-9) as a forward operator in combination with a soil hydraulic model. Similar approaches have been conducted using e.g. the COSMIC forward operator model code (Shuttleworth et al., 2013) for above-ground CRNS applications (e.g., Brunetti et al., 2019; Barbosa et al., 2021). Although the application as a forward operator in combination with soil hydraulic modelling may produce more accurate results as the soil water transport is simulated in different depths and also allows for the retrieval of soil moisture simulated in several soil layers, a large number of input parameters is required which may not be available at all sites. Furthermore, coupling the derived equations with a soil hydraulic model may introduce additional uncertainties by the model assumptions and by the propagation of uncertainties from input parameters."

**Additional references:**

Gudima, K. K., Mashnik, S. G., Toneev, V. D.: Cascade-exciton model of nuclear reactions, Nuclear Physics A, 401, 329-361, https://doi.org/10.1016/0375-9474(83)90532-8, 1983.

Heusser, G.: Cosmic ray interaction study with low-level Ge-spectrometry. Nuclear Instruments and Methods in Physics Research A, 369, 539-543, https://doi.org/10.1016/S0168-9002(96)80046-5, 1996.

Kodama, M., Kudo, S., and Kosuge, T.: Application of atmospheric neutrons to soil moisture measurement, Soil
Science, 140, 237–242, 1985.

Mollerach, A., Roulet, E.: Progress in high-energy cosmic-ray physics, Progress in Particle and Nuclear Physics, 98, 85-118, https://doi.org/10.1016/j.ppnp.2017.10.002, 2018.

**Additional changes in the revised manuscript:**

**From line 97:**

Original:

"From a depth of 850 cm a layer of glacio-fluvial coarse sands containing fine gravel components which extents downward until reaching the glacial till deposited during the earlier Frankfurt phase of the Weichselian glaciation."

Adjustment:

"From a depth of 850 cm a layer of glacio-fluvial coarse sands containing fine gravel components which extends downward until reaching the glacial till deposited during an earlier phase of the Weichselian glaciation."

**From line 264:**

Original:

"The additional subset of simulations revealed that neutron ratios for a well tube composed of PVC are on average 60 % lower compared to a stainless steel tube with equal wall thickness but respond similarly to changes in soil moisture. In addition, a PVC material with a thickness of 7.5 mm produces neutron ratios which are on average 28 % lower compared to a thinner PVC tubing with a wall thickness of only 5 mm."

Adjustment:

"The additional subset of simulations revealed that neutron ratios for a well tube composed of stainless steel are on average 60 % higher compared to a PVC tube with equal wall thickness but respond similarly to changes in soil moisture. In addition, a thinner PVC material with a thickness of only 5 mm produces neutron ratios which are on average 28 % higher compared to a PVC tubing with a wall thickness of 7.5 mm."